# Design Linear Constrained Neural Layers with Implicit Convex Optimization

**Junchi Yan** [1]  **Jiaxi Liu** [1]  **Yihui Tu** [2]  **Fangyuan Zhou** [1]  **Wenzheng Pan** [1]  **Zhongteng Gui** [1]  **Liangliang Shi** [1][3]

## Abstract

One essential limitation of neural networks is how to enforce (hard) constraints on prediction. We propose a plug-in, differentiable layer, which involves a fast implicit (convex) optimization procedure to enforce the general linear constraint. It aims to minimize a divergence between unconstrained and constrained outputs. Connecting to and beyond existing handcrafted layers, we show that our layer degrades to classic layers like Softmax, Sinkhorn and tanh etc. when the corresponding constraint is enforced by KL-divergence minimization. We further show that by replacing the KL-div with a Euclidean distance, a closed-form solution can be derived for highly-efficient constraint enforcing. We evaluate the above two variants of layers, termed as BLCLayer and GLCLayer, with their corresponding neural solver BLCNet and GLCNet with simple MLP/GNN-like backbone. Experiments on linear programming, as well as two real-world problems: partial graph matching and portfolio allocation which involve other discrete constraints.

## 1. Introduction

While deep neural networks have excelled in classification and regression, recent advances have expanded their application to diverse domains where satisfying constraints is often mandatory (Bengio et al., 2021). It remains an open problem (Wang et al., 2023a) and calls for more general layers that can fulfill diversified constraint requirement.

One approach to ensure that the network predictions satisfy certain constraints is via reinforcement learning (RL) (Liu et al., 2024). RL has the capacity to handle very broad

[1]Sch. of Artificial Intelligence and Sch. of Computer Science, Shanghai Jiao Tong University [2]Department of Mathematics, Shanghai University [3]Shanghai Institute for Mathematics and Interdisciplinary Sciences. Correspondence to: Junchi Yan <yanjnchi@sjtu.edu.cn>, Liangliang Shi <shil021@simis.cn>, Yihui Tu <tuyihui@shu.edu.cn>.

*Proceedings of the 43rd International Conference on Machine Learning*, Seoul, South Korea. PMLR 306, 2026. Copyright 2026 by the author(s).

classes of constraints in an auto-regressive manner. However, it often faces issues e.g. large action spaces, sparse reward functions, and computationally expensive training. More importantly, it cannot always generalize well to new instances especially in safety-critical or high-stakes scenarios, such as chip design (Cheng & Yan, 2021). Another line of works e.g., DIFUSCO (Sun & Yang, 2023) resort to state-of-the-art training scheme e.g., diffusion models. It is expected that constraints are implicitly satisfied during training. However, there is no theoretical guarantee and lacks an explicit scheme to control the output behavior. On the other hand, tailored neural nets are designed for constraints like permutation (Wang et al., 2023b) and partial permutation (Wang et al., 2023a). Yet a general scheme capable of handling arbitrary linear constraints is still lacking.

On the other hand, linear constraints are powerful and widely used across areas, and a pertinent example is linear programming (LP) where both constraints and objective are linear w.r.t. the variables. Recently there are emerging efforts in dealing with certain linear constraints inside a neural network via a tailored layer. For example, a special yet important type of linear constraints, i.e., positive linear constraint is attained by altering the Sinkhorn algorithm with the resulting new layer in the seminal work LinSATNet (Wang et al., 2023c). As a preliminary technique, it yet suffers computational inefficiency and sometimes convergence instability. An improvement is made in GLinSAT (Zeng et al., 2024), which introduces a differentiable projection layer via entropy-regularized LP. It advances the field by supporting general linear constraints, enabling GPU-accelerated batch processing without matrix factorization, and providing theoretical convergence guarantees via accelerated gradient descent. However, it incurs new practical challenges: Its accelerated gradient descent mechanism requires careful tuning of step-size hyperparameters that critically influence convergence speed and solution quality.

In this paper, we introduce LinConLayer (i.e., LCLayer for abbreviation), which designs the **Linear Constrained Layers** by minimizing the divergence between the layer's output and its input. We first demonstrate that classic layers, such as Softmax, Sigmoid, and Tanh, can be derived as solutions to implicit optimization problems that minimize KL-divergence under specific constraints. We then generalize these special layers to the general case un-

*Table 1.* Constraint enforcing layers. Note the GLinSAT involves gradient descent which is nontrivial especially when itself is an inner loop inside an outer loop of gradient descent i.e., network training. In contrast, our method is highly efficient and stable (e.g., see Table 5).

| Methods | output x range | constraint form | algorithm | efficiency |
|---|---|---|---|---|
| LinSAT (Wang et al., 2023c) | [0, 1] | positive linear | generalized Sinkhorn | low |
| GLinSAT (Zeng et al., 2024) | $[0, \infty]$ | linear | gradient descent | medium |
| HardNet (Min & Azizan, 2024) | $[-\infty, \infty]$ | Input-dependent affine | Closed-form via Pseudo-inv | medium |
| BLCLayer (Sec. 2.2) | $[0, \infty]$ | binary positive linear | Bregman projection | high |
| GLCLayer (Sec. 2.3) | $[-\infty, \infty]$ | linear | analytical solution | high |

der the KL-divergence minimization framework and call the resulting layer as: Binary Linear Constrained Layers (BLCLayer), where the coefficients of the linear constraints are assumed binary. We adopt Bregman Iterative Projections algorithms (Benamou et al., 2015), with sound convergence properties, to solve the implicit optimization problem. For general linear constraints, $\mathbf{W}\mathbf{x} = \mathbf{b}$, we replace the KL-divergence with Euclidean distance and derive a closed-form solution in each iteration without using gradient descent, to obtain the new general linear constraint layer (GLCLayer). Compared with (Wang et al., 2023c), our GLCLayer no longer requires the coefficient matrices to be positive, and the final output entries can be arbitrary real numbers – see Table 1 for a comparison for the related methods. The pipeline is shown in Fig. 1 with the highlights as follows.

1) We propose designing constraint-satisfying networks using implicit optimization, in which fulfilling the output layer can be seen as solving a constrained convex optimization problem via differentiable matrix iteration. Our framework can incorporate existing layers like Softmax and Sinkhorn layers, as viewed as special cases w.r.t. certain constraints, when the objective involves KL divergence.

2) We extend Softmax and Sinkhorn layers by modifying their original constraints to binary positive linear constraints (BLC), leading to a new layer namely BLCLayer, whose implicit optimization can be efficiently solved by Bregman iterative projection without gradient descent. Experiments on the Partial Graph Matching and Portfolio Allocation tasks show that BLCNet (adding a backbone with our devised BLCLayer) performs much more stable and efficient than the tailored SOTA LinSATNet (Wang et al., 2023c).

3) For the more general linear constraints (GLC), we devise the corresponding GLCLayer, achieved by replacing the KL divergence with Euclidean distance as the objective of implicit optimization. Based on GLCLayer, we design a neural solver with a neural backbone for linear programming, which shows high efficiency especially on large instances.

**Notation.** In line with Fig. 1, we define $\mathbf{y} \in \mathbb{R}^n$ as the unconstrained output of a network and $\mathbf{x} \in \mathbb{R}^n$ as its constrained one. It also holds in matrix form i.e., $\mathbf{Y}, \mathbf{X} \in \mathbb{R}^{n \times m}$ e.g., in graph matching.

**Preliminaries.** We discuss the background: Optimal Trans-

port and Iterative Bregman Projections.

**Optimal Transport.** We first introduce Optimal Transport (OT) (Monge, 1781), which aims to find a mapping that minimizes the total cost of transporting mass from a source measure to a target measure. Kantorovich's formulation (Kantorovich, 1942) extends the original problem by using probabilistic transport, which is now the standard approach. Given a cost matrix $\mathbf{C} \in \mathbb{R}_+^{m \times n}$ and histograms $(\mathbf{a}, \mathbf{b})$, where $m$ and $n$ represent the dimensions, the entropic regularized Kantorovich OT problem is formulated as:

$$\min_{\mathbf{X} \in U(\mathbf{a}, \mathbf{b})} \langle \mathbf{C}, \mathbf{X} \rangle - \epsilon H(\mathbf{X}), \quad (1)$$

where $U(\mathbf{a}, \mathbf{b}) = \{\mathbf{X} \in \mathbb{R}_{mn}^+ \mid \mathbf{X}\mathbf{1}_n = \mathbf{a}, \mathbf{X}^\top \mathbf{1}_m = \mathbf{b}\}$ and $\epsilon > 0$ is the entropic regularization parameter. The entropy term $H(\mathbf{X}) = -\langle \mathbf{X}, \log \mathbf{X} - \mathbf{1}_{m \times n} \rangle$ enforces smoothness in the transport plan $\mathbf{X}$, which leads to a strongly convex objective, ensuring a unique solution that takes the form $\mathbf{X}_\epsilon^* = \text{diag}(\mathbf{u})\mathbf{K}\text{diag}(\mathbf{v})$, where $\mathbf{K} = e^{-\mathbf{C}/\epsilon}$ is the Gibbs kernel with scaling variables $(\mathbf{u}, \mathbf{v})$ (Cuturi, 2013).

**Iterative Bregman Projections** Bregman Iterative Projections provide an efficient way to solve convex optimization problems, including entropic optimal transport (Benamou et al., 2015). Specifically, given a positive matrix $\mathbf{K}$, it optimizes $\min_{\mathbf{X} \in \mathcal{C}} \text{KL}(\mathbf{X} \| \mathbf{K})$, where $\mathcal{C} = \bigcap_{l=1}^L \mathcal{C}_l$ is a convex set representing the constraints on $\mathbf{X}$. Here, we assume that $\mathcal{C}$ can be expressed as the intersection of $L$ convex sets, and the minimization is performed by iterating over KL projections onto each constraint set $\mathcal{C}_l$. Starting from $\mathbf{X}^{(0)} = \mathbf{K}$, the iterations proceed as:

$$\mathbf{X}^{(l)} = \text{Proj}_{\mathcal{C}_n}^{\text{KL}}(\mathbf{X}^{(l-1)}) = \arg\min_{\mathbf{X} \in \mathcal{C}_l} \text{KL}(\mathbf{X} \| \mathbf{X}^{(l-1)}), \quad (2)$$

As proven by Bregman (Bregman, 1967), these projections converge to the optimal solution of the optimization problem as $n \to \infty$, yielding the solution:

$$\mathbf{X} = \text{Proj}_{\mathcal{C}}^{\text{KL}}(\mathbf{K}) = \lim_{n \to \infty} \text{Proj}_{\mathcal{C}_n}^{\text{KL}}(\mathbf{X}^{(n-1)}). \quad (3)$$

This method is commonly applied to solve entropic optimal transport problems and variants of OT when the cost matrix $\mathbf{C} \in \mathbb{R}_{nn}^+$ is used with the Gibbs kernel $\mathbf{K} = e^{-\mathbf{C}/\epsilon}$, and the constraints correspond to the marginal constraints in OT.

The Sinkhorn algorithm (Sinkhorn, 1964), which is based on the iterative normalization of rows and columns of a matrix, is a specific application of the Bregman Iterative Projections. For example, in image matching and permutation problems, we set $\mathbf{X}^{(0)} = \mathbf{K}$ and $\mathcal{C} = \mathcal{C}_1 \bigcap \mathcal{C}_2$ where $\mathcal{C}_1 = \{\mathbf{X}|\mathbf{X}\mathbf{1} = \mathbf{1}\}$ and $\mathcal{C}_2 = \{\mathbf{X}|\mathbf{X}^\top\mathbf{1} = \mathbf{1}\}$, then according to Eq. 2 and Eq. 3, we can perform the Sinkhorn iterations as:

$$\begin{aligned}
\mathbf{X}^{(2k+1)} &= \mathbf{X}^{(2k)} \oslash \left(\mathbf{X}^{(2k)}\mathbf{1}_{n\times n}\right), \\
\mathbf{X}^{(2k+2)} &= \mathbf{X}^{(2k+1)} \oslash \left(\mathbf{1}_{n\times n}\mathbf{X}^{(2k+1)}\right),
\end{aligned} \quad (4)$$

where $\oslash$ denotes element-wise division, and $\mathbf{1}_{n\times n}$ is the all-one matrix. This iterative procedure enforces doubly stochastic constraints and can be viewed as a Bregman projection applied to entropic optimal transport. It is commonly utilized as a layer in neural networks for tasks such as visual matching (Wang et al., 2023b), ranking (Mena et al., 2018), and others (Sander et al., 2022).

## 2. LinConNet: Linear Constraint via Implicit Optimization Design

For clarity, we begin by reconsidering the classic layer from the perspective of KL divergence-based implicit optimization, and then modify the constraints to extend it to general cases. Specifically, given raw data $\mathbf{x}$ and a neural network $f_\theta(\cdot)$, one can obtain the unconstrained output vector $\mathbf{y}$. In this paper, our core task is to design a network such that the final output of the neural network satisfies (linear) constraints, such as $\mathbf{W}\mathbf{x} = \mathbf{b}$, where $(\mathbf{W}, \mathbf{b})$ are known.

### 2.1. Implicit Optimization View to Classic Constraint Layers: Softmax and Tanh

Inspired by the fact that the Sinkhorn Layer can be derived from constrained KL divergence, we propose that networks can be designed from the perspective of implicit optimization. First, we revisit several traditional layers (e.g., Softmax, Sigmoid, and Tanh) through the lens of implicit optimization and find that these networks can be viewed as special cases under constrained KL divergence.

Now we show that Softmax can also be understood from the implicit optimization view based on KL divergence. Given an unconstrained output $\mathbf{y}$, we consider the implicit optimization under constraint $\mathcal{C} = \{\mathbf{x} \geq \mathbf{0}|\mathbf{x}^\top\mathbf{1} = 1\}$:

$$\min_{\mathbf{x}\geq\mathbf{0}} \mathrm{KL}(\mathbf{x}|e^{\mathbf{y}/\epsilon}) \quad \text{s.t.} \quad \mathbf{x}^\top\mathbf{1} = 1, \quad (5)$$

Through the method of Lagrange multipliers, the above problem has a closed-form solution $\mathbf{x} = \frac{e^{\mathbf{y}/\epsilon}}{\sum_i e^{\mathbf{y}_i/\epsilon}}$, which is exactly the Softmax operator. Therefore, Softmax can be viewed as an activation layer derived from an implicit optimization involving KL divergence, with the constraint

*Table 2.* Implicit optimization perspective for Softmax, Sigmoid, Tanh layers, given the unconstrained output $\mathbf{y}$ or $\mathbf{Y}$ of networks. Note that for BLC, it involves Bregman iteration while GLC requires matrix inverse to obtain the analytical solution.

| Layer | Formula | Optimization Objective | Constraints |
|---|---|---|---|
| Softmax | $\mathbf{x} = \frac{e^{\mathbf{y}/\epsilon}}{\sum_i e^{\mathbf{y}_i/\epsilon}}$ | $\min_{\mathbf{x}\geq\mathbf{0}} \mathrm{KL}(\mathbf{x}\|e^{\mathbf{y}/\epsilon})$ | $\mathbf{x}^\top\mathbf{1} = 1$ |
| Sigmoid | $\mathbf{x} = \frac{1}{1+e^{-\mathbf{y}}}$ | $\min_{\mathbf{x}} \mathrm{KL}\left([\mathbf{x}, 1-\mathbf{x}] \| [e^{\mathbf{y}/2}, e^{-\mathbf{y}/2}]\right)$ | $\mathbf{x} \in (0, 1)$ |
| Tanh | $\mathbf{x} = \frac{e^{\mathbf{y}}-e^{-\mathbf{y}}}{e^{\mathbf{y}}+e^{-\mathbf{y}}}$ | $\min_{\mathbf{x}} \mathrm{KL}\left(\left[\frac{\mathbf{x}+1}{2}, \frac{1-\mathbf{x}}{2}\right] \| [e^{\mathbf{y}}, e^{-\mathbf{y}}]\right)$ | $\mathbf{x} \in (-1, 1)$ |
| Sinkhorn | See Eq. 4 | $\min_{\mathbf{X}} \mathrm{KL}\left(\mathbf{X} \| \exp(\mathbf{Y}/\epsilon)\right)$ | $\mathbf{X}\mathbf{1} = \mathbf{1}, \mathbf{X}^\top\mathbf{1} = \mathbf{1}$ |
| BLCLayer | See Eq. 9 | $\min_{\mathbf{x}\geq\mathbf{0}} \mathrm{KL}(\mathbf{x}\|e^{\mathbf{y}/\epsilon})$ | $\mathbf{W}\mathbf{x} = \mathbf{b}$ |
| GLCLayer | See Eq. 12 | $\min_{\mathbf{x}} \frac{1}{2}\|\mathbf{x} - \mathbf{y}\|^2$ | $\mathbf{W}\mathbf{x} = \mathbf{b}$ |

$\mathbf{x}^\top\mathbf{1} = 1$. Similarly, other layers, such as Sigmoid, can be reconsidered from the perspective of KL divergence. Unlike the constraints on matrix or vector summation seen in Sinkhorn or Softmax, the constraint for Sigmoid is restricted to $\mathbf{x} \in (0, 1)$. We construct this implicit optimization by defining the variable $1 - \mathbf{x}$, which can be formulated as:

$$\min_{\mathbf{x}} \mathrm{KL}\left([\mathbf{x}, 1-\mathbf{x}] \| \left[e^{\mathbf{y}/2}, e^{-\mathbf{y}/2}\right]\right) \quad \text{s.t.} \quad \mathbf{x} \in (0, 1). \quad (6)$$

The optimization problem seeks to align the probability vector $[\mathbf{x}, 1-\mathbf{x}]$ with the feature vector $[e^{\mathbf{y}/2}, e^{-\mathbf{y}/2}]$, ensuring that $\mathbf{x}$ remains within a valid probability range. This can also be solved using the method of Lagrange multipliers. In addition to the Sigmoid layer, the Tanh layer can also be understood from the perspective of KL divergence. A summary of this is presented in Table 2, with the proofs provided in Appendix B.3 and B.5.

### 2.2. BLCLayer: Binary Linear Constraint Layer via Bregman Projections

By KL-divergence, we now demonstrate how to generalize the Softmax and Sinkhorn layers to new layers that accommodate different constraints. In this paper, we focus on the linear constraint $\mathbf{W}\mathbf{x} = \mathbf{b}$ and propose the Binary Linear Constrained Layer (BLCLayer). Our key idea is that, since the Sinkhorn layer satisfies row and column sum constraints through Bregman iterations, it raises the question of whether it can be extended to handle more complex and general constraints. Initially, we consider the case where the constraint matrix $\mathbf{W} \in \{0, 1\}^{m\times n}$ is binary, and aim to solve:

$$\min_{\mathbf{x}\geq\mathbf{0}} \mathrm{KL}(\mathbf{x} \mid e^{\mathbf{y}}) \quad \text{s.t.} \quad \mathbf{W}\mathbf{x} = \mathbf{b}. \quad (7)$$

Unlike Softmax/Sigmoid/Tanh layers, having explicit solutions, the above implicit optimization has no closed-form solution. We need to adopt an approach similar to the Sinkhorn algorithm, using Bregman iterative projections to decompose the constraints and obtain its iterative solution. We decompose the constraint set $\mathcal{C} = \{\mathbf{x} \geq \mathbf{0} \mid \mathbf{W}\mathbf{x} = \mathbf{b}\}$ into $m$ subsets of constraints: $\mathcal{C} = \bigcap_{l=1}^{L} \mathcal{C}_l$, where $\mathcal{C}_l = \{\mathbf{x} \geq \mathbf{0} \mid \mathbf{w}_{(l)}^\top\mathbf{x} = b_{(l)}\}$, and we define $\mathcal{C}_l = \mathcal{C}_{l+m}$ for $l < m$. By applying Bregman iterative projections, we

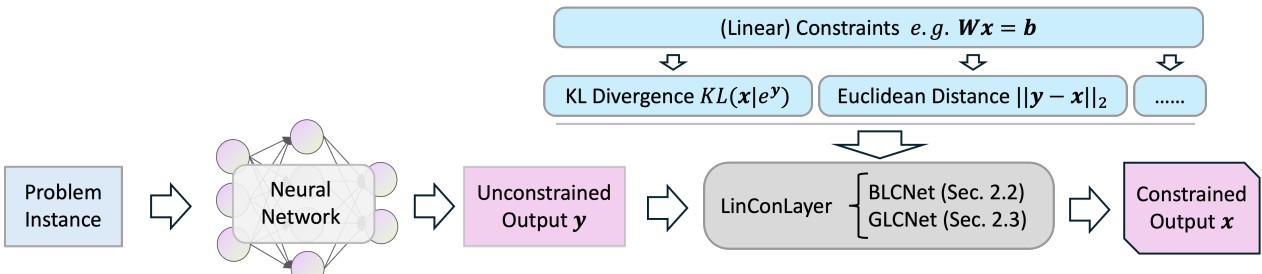

*Figure 1.* Pipeline: i) binary positive linear constraint version (BLC) and ii) general version (GLC).

solve it iteratively by optimizing the subproblems of Eq. 7:

$$\mathbf{x}^{(l+1)} = \arg\min_{\mathbf{x}\geq\mathbf{0}} \mathrm{KL}(\mathbf{x} \mid \mathbf{x}^{(l)}) \quad \text{s.t.} \quad \mathbf{w}_{(l)}^{\top}\mathbf{x} = b_{(l)} \quad (8)$$

where $\mathbf{w}_{(l)}$ and $b_{(l)}$ are the $l$-th components of the matrix $\mathbf{W}^{\top}$ and the vector $\mathbf{b}$, respectively, and we define $\mathbf{w}_{(l+m)} = \mathbf{w}_{(l)}$, $\mathbf{b}_{(l+m)} = b_{(l)}$ for $l > 0$. Unless otherwise specified, subscripts denote dimension indices, and superscripts indicate iteration steps. Vectors and matrices are typeset in bold, and scalar values in regular type. By initializing $\mathbf{x}^{(0)} = e^{\mathbf{y}}$, and iterating $l$ from 0 to a sufficiently large number, we can obtain Eq. 7 by solving Eq. 8. Then, to solve Eq. 8, since $\mathbf{w} \in \{0, 1\}^n$, we can obtain the explicit solution to the optimization in Eq. 8 as follows:

$$\mathbf{x}^{(l+1)} = (\mathbf{1}_n - \mathbf{w}_{(l)}) \odot \mathbf{x}^{(l)} + \mathbf{w}_{(l)} \odot \frac{b_{(l)}}{\mathbf{w}_{(l)}^{\top}\mathbf{x}^{(l)}}\mathbf{x}^{(l)} \quad (9)$$

We provide the derivation in Appendix B.6, and a convergence proof for the iteration described by Eq. 8 is provided in Appendix E. The above iteration, based on vector multiplication and division, corresponds to the BLCLayer for the linear constraint $\mathbf{Wx} = \mathbf{b}$.

Furthermore, consider the case of inequality constraints (e.g., $\mathbf{w}_{(l)}^{\top}\mathbf{x} \leq b_{(l)}$). In (Zeng et al., 2024), it converts the inequality into an equality constraint by introducing additional positive learnable parameters. Alternatively, the iteration can be adjusted as already done for partial Optimal Transport (OT). By replacing the constraint $\mathbf{w}_{(l)}^{\top}\mathbf{x}$ in Eq. 8 with the inequality $\mathbf{w}_{(l)}^{\top}\mathbf{x} \leq b_{(l)}$, we can update the iteration:

$$\mathbf{x}^{(l+1)} = (\mathbf{1}_n - \mathbf{w}_{(l)}) \odot \mathbf{x}^{(l)} + \min\left\{1, \frac{b_{(l)}}{\mathbf{w}_{(l)}^{\top}\mathbf{x}^{(l)}}\right\} \cdot \mathbf{w}_{(l)} \odot \mathbf{x}^{(l)}. \quad (10)$$

For the constraint $\mathbf{w}_{(l)}^{\top}\mathbf{x} = b_{(l)}$, one only needs to replace the min operation with a max operation in Eq. 10." In practice, although the assumption of a matrix $\mathbf{W} \in \{0, 1\}^{m \times n}$ may appear restrictive, it is sufficiently general to handle most graph-related scenarios. The experiments in Sec. 3.1 and Sec. 3.2 provide two examples that demonstrate its capabilities. Of course, the BLCLayer does have some limitations, such as the requirement for positive outputs and the

constraint coefficients being binary, in order to ensure explicit solutions for the subproblems. In next subsection, we discuss how to replace the KL divergence with alternative methods to accommodate more general constraints.

### 2.3. GLCLayer: General Linear Constraint Layer via Implicit Optimization

Recall that the above BLCLayer has limitations as it handles only constraints where the variables are positive and its constraint parameters $\mathbf{W} \in \{0, 1\}^{n \times m}$. For a more general satisfaction, we propose a general linear constraint layer designed for seamless integration into neural architectures.

**Network Design for General Linear Satisfaction $\mathbf{Wx} = \mathbf{b}$.** Given the equality constraints[1] $\mathbf{Wx} = \mathbf{b}$, we assume $m < n, rank(W) = m$ and there exist a feasible solution $\mathbf{x}_0$ that satisfying $\mathbf{Wx}_0 = \mathbf{b}$. Following the previous discussion, if we consider the KL divergence as the optimal implicit optimization objective function, there are two main issues: 1) The output $\mathbf{x}$ is necessarily non-negative, which may not be suitable for certain cases, such as regression problems, where negative values for $\mathbf{x}$ are allowed; and 2) Solving Eq. 7 or Eq. 8 inevitably introduces additional Lagrange multipliers, which are not conducive to backpropagation during the training process. A natural progression is to explore alternative objective functions, such as the Euclidean distance, to derive a simpler layer that satisfies the constraints. Thus, we attempt to replace the KL divergence with the Euclidean distance for implicit optimization as:

$$\min_{\mathbf{x}} \frac{1}{2}||\mathbf{x} - \mathbf{y}||^2 \quad \text{s.t.} \quad \mathbf{Wx} = \mathbf{b} \quad (11)$$

where $\mathbf{y}$ is the unconstrained output and $\mathbf{x}$ is the constrained output satisfying $\mathbf{Wx} = \mathbf{b}$. Thus, we can easily obtain the solution for the above implicit optimization:

$$\mathbf{x} = \mathbf{y} - \mathbf{W}^{\top}(\mathbf{W}\mathbf{W}^{\top})^{-1}(\mathbf{W}\mathbf{y} - \mathbf{b}) \quad (12)$$

The derivation details are provided in Appendix B.2. Note that $(\mathbf{W}, \mathbf{b})$ are known matrices or vectors, and therefore

---

[1]Inequality constraints can be transformed to equality in the design of networks as discussed in (Zeng et al., 2024).

the training process does not involve computing gradients with respect to $(\mathbf{W}, \mathbf{b})$. Additionally, for this GLCLayer, we can compute the following:

$$\frac{\partial \mathcal{L}}{\partial \mathbf{y}} = \frac{\partial \mathcal{L}}{\partial \mathbf{x}} \frac{\partial \mathbf{x}}{\partial \mathbf{y}} = (\mathbf{I} - \mathbf{W}^\top (\mathbf{W}\mathbf{W}^\top)^{-1} \mathbf{W}) \frac{\partial \mathcal{L}}{\partial \mathbf{x}}. \quad (13)$$

Thus, ideally, the above solution can be obtained without iterations. Compared to other approaches for satisfying linear constraints $\mathbf{W}\mathbf{x} = \mathbf{b}$, GLCLayer requires no additional hyperparameters (e.g., Lagrange multipliers as often used in literature), which can affect gradient backpropagation and make the convergence of iterative algorithms difficult to control when implemented as a layer. The main limitation of this method, however, is that it requires computing the inverse of $\mathbf{W}\mathbf{W}^\top$ via an LU decomposition, which must be precomputed before the training and inference processes. It can also incur numerical instability.

**Special Cases in $\mathbf{W}\mathbf{x} = \mathbf{b}, \mathbf{x} \geq 0$ for Linear Programming.** For the linear constraint $\mathbf{W}\mathbf{x} = \mathbf{b}$, we naturally associate it with the linear programming problem: $\min_\mathbf{x} \mathbf{c}^\top \mathbf{x}$ s.t. $\mathbf{W}\mathbf{x} = \mathbf{b}, \mathbf{x} \geq 0$. In this context, we explore the possibility of optimizing this linear programming problem using a neural network that is linear and feasible. Clearly, while the constraints $\mathbf{W}\mathbf{x} = \mathbf{b}, \mathbf{x} \geq 0$ are linear, they can be reformulated as $\tilde{\mathbf{W}}\mathbf{x} = \mathbf{b}$. However, the matrix $\tilde{\mathbf{W}}$ corresponding to these modified constraints will always have fewer columns than rows, i.e., $n < m$, which implies that $\tilde{\mathbf{W}}\tilde{\mathbf{W}}^\top$ does not exist. In this case, we are forced to introduce additional variables to design the network. For the LP problem, we modify the implicit optimization:

$$\min_\mathbf{x} \mathbf{c}^\top \mathbf{x} + \frac{1}{2}\|\mathbf{x} - \mathbf{y}\|^2 \quad \text{s.t.} \quad \mathbf{W}\mathbf{x} = \mathbf{b}, \mathbf{x} \geq 0 \quad (14)$$

We can solve the above problem using the ADMM method. Specifically, by introducing the variable $\mathbf{z}$ with constraints $\mathbf{z} = \mathbf{x}$, the Lagrange multiplier $\lambda$, and the penalty parameter $\rho$, we can formulate the Lagrangian function:

$$L(\mathbf{x}, \mathbf{z}, \lambda) = \mathbf{c}^\top \mathbf{x} + \frac{1}{2}\|\mathbf{x} - \mathbf{y}\|^2 + \lambda^\top (\mathbf{W}\mathbf{x} - \mathbf{b}) + \rho\|\mathbf{x} - \mathbf{z}\|^2, \quad (15)$$

then iteration can be updated as

$$\mathbf{x}^{(k+1)} = \frac{\mathbf{y} - \mathbf{c} + \mathbf{W}^\top \lambda^{(k)}}{1 + 2\rho\mathbf{y}}, \quad \mathbf{z}^{(k+1)} = \max\{0, \mathbf{x}^{(k+1)}\},$$

$$\lambda^{(k+1)} = \lambda^{(k)} + (\mathbf{W}\mathbf{x}^{(k+1)} - \mathbf{b}), \quad (16)$$

where $k$ represents the iteration number. In the network design, we initialize $\mathbf{x}^{(0)} = \mathbf{y}$ and $\lambda^{(0)} = \mathbf{0}$, and then iteratively update based on Eq. 16 until convergence. From an optimization perspective, our layer design combines the parallel iterative algorithm of optimization with the prediction of the neural network. It simplifies the techniques traditional algorithm for convergence by learning a proximal point; on the other hand, by supervised or unsupervised training, it leverages the neural network to accelerate inference rapidly.

*Table 3.* Mean F1, train and inference time of graph matching on Pascal VOC Keypoint (unfiltered setting).

| GM Net | Constraint Technique | F1 ↑ | Train ↓ | Test ↓ |
|---|---|---|---|---|
| BBGM | differentiable blackbox (Pogančić et al., 2020) | 0.5516 | 1h 12m | 1h 21m |
| NGMv2 | Sinkhorn | 0.5453 | 1h 25m | 1h 34m |
| NGMv2 | LinSAT-5 iters | 0.5858 | 1h 56m | 2h 15m |
| NGMv2 | LinSAT-10 iters | 0.5839 | 2h 19m | 2h 37m |
| NGMv2 | BLCLayer-5 iters (ours) | 0.5972 | 1h 28m | 1h 36m |
| NGMv2 | BLCLayer-10 iters (ours) | 0.6012 | 1h 37m | 1h 46m |

## 3. Experiments

In line with (Wang et al., 2023c) on positive linear constraint, we apply BLCLayer as a plug-in to the shared backbones across baseline methods, with two real-world tasks: partial graph matching and portfolio allocation in Sec. 3.1 and 3.2. In Sec. 3.3, we introduce a new task: learning LP with the proposed GLCLayer and further exploration. All experiments are implemented by PyTorch 2.5.1 on dual Intel(R) 8352V CPUs and an NV 4090 GPU (24GB). We integrate the layers into a backbone network and term the whole network solver as BLCNet and GLCNet (see Table 4).

### 3.1. Experiments on Visual Graph Matching

**Problem.** Partial GM refers to the setting when the matching graphs contain different numbers of nodes. We follow (Wang et al., 2023a) by formulating it as a top-k selection task with an estimated number of inliers and devised a top-k module plugged into deep graph matching pipelines.

**Constraint Formulation and Layer Design.** For graphs $G_1 = (V_1, E_1)$ and $G_2 = (V_2, E_2)$ with cardinalities $|V_1| = n_1$ and $|V_2| = n_2$. $M \in \mathbb{R}^{n_1 \times n_2}$ is used to represent the potential correspondences between nodes in $G_1$ and $G_2$, with $M_{i,j}$ indicating the matching score between node $i$ in $G_1$ and node $j$ in $G_2$. In previous bijective graph matching networks, the one-to-one node matching constraint is enforced by the Sinkhorn algorithm. It formulates the matching problem by adding a partial matching constraint: assume that the number of matchable inliers is $\phi$:

$$\mathbf{X}\mathbf{1}_{n_2} \leq \mathbf{1}_{n_1}, \quad \mathbf{X}^\top \mathbf{1}_{n_1} \leq \mathbf{1}_{n_2},$$
$$\mathbf{1}_{n_1}^\top \mathbf{X}\mathbf{1}_{n_2} = \phi, \quad \mathbf{X} \in \{0,1\}^{n_1 \times n_2} \quad (17)$$

Then with the procedure in Sec. 2.1, we get the iterations with the projections ($\mathbf{X}$ omitted):

$$\text{Proj}_{\mathcal{C}_1}^{KL} = \text{diag}(\min\left(\mathbf{1}_{n_1}, \frac{\mathbf{1}_{n_1}}{\mathbf{X}\mathbf{1}_{n_2}}\right))\mathbf{X},$$

$$\text{Proj}_{\mathcal{C}_2}^{KL} = \mathbf{X}\text{diag}(\min\left(\frac{\mathbf{1}_{n_2}}{\mathbf{X}^\top \mathbf{1}_{n_1}}, \mathbf{1}_{n_2}\right)), \quad (18)$$

$$\text{Proj}_{\mathcal{C}_3}^{KL} = \frac{\phi\mathbf{X}}{\mathbf{1}_{n_1}^\top \mathbf{X}\mathbf{1}_{n_2}}.$$

Finally, as introduced by (Benamou et al., 2015), assuming $\mathcal{C}_l = \mathcal{C}_{l+3}$ with positive integer $l$ as the index of Bregman

*Table 4.* Sharpe Ratio and Runtime (in seconds) for Portfolio Allocation. Our methods achieve SOTA with much faster inference.

| Method | Sharpe ↑ | Train (s) ↓ | Inf. (s) ↓ |
|---|---|---|---|
| Unconstrained | 2.190 | **85.88** | 8.99 |
| LinSAT-10 | 2.508 | 976.08 | 73.00 |
| LinSAT-100 | 2.755 | 4862.5 | 524.39 |
| cvxpylayers | 3.058 | 271.70 | 26.66 |
| BLCNet-5 (Ours) | 2.773 | 202.65 | 14.34 |
| BLCNet-10 (Ours) | 2.770 | 268.21 | 18.85 |
| GLCNet (Ours) | **3.053** | 91.86 | **7.20** |

*Table 5.* Total solving time, Bregman iteration time, and iteration cost ratio with varying problem scales. We set batch size to 8 and iteration number to 5 to align with previous settings in Table 3.

| Max Scale | Total Time | Bregman Time | Bregman Ratio |
|---|---|---|---|
| 10 | 0.1573 | 0.0119 | 7.57% |
| 20 | 0.1786 | 0.0119 | 6.68% |
| 40 | 0.2009 | 0.0119 | 5.93% |
| 80 | 0.2244 | 0.0119 | 5.31% |
| Unlimited | 0.2707 | 0.0117 | 4.32% |

iteration, the minimization is solved with iteration: $\mathbf{X}^{(n)} = \text{Proj}_{\mathcal{C}_n}^{KL}\left(\mathbf{X}^{(n-1)}\right)$, starting from $\mathbf{X}^{(0)} = \mathbf{K} = e^{\mathbf{Y}/\epsilon}$. We can get the solution by $\mathbf{X}^* = \lim_{n\to\infty} \text{Proj}_{\mathcal{C}_n}^{KL}(\mathbf{X}^{(n-1)})$, which performs the iterations for Eq. 2 and (Bregman, 1967) guarantees its convergence.

**Experiment Details.** We follow NGMv2 (Wang et al., 2023b) as backbone, and replace the Sinkhorn and LinSAT layer with BLCLayer. In NGMv2, a VGG16 (Qassim et al., 2018) is used to extract node and global features, refined by SplineConv (Fey et al., 2018), while the edge features are derived from the node features and graph connections. Matching scores are further predicted using the neural graph matching network (Wang et al., 2023b), resulting in a matrix denoting matching prediction. This matrix is then input into our BLCLayer to enforce constraints from Eq. 17. Its output is reshaped into matrix for end-to-end training with permutation loss. During inference, the Hungarian algorithm is applied to the matching matrix to keep only the highest matching scores and discard the rest of the unpaired nodes.

The baselines: BBGM (Rolínek et al., 2020), Sinkhorn, LinSAT, and BLCNet share the same training configurations. We use a batch size 8 and train for 10 epochs, with each epoch of 1K iterations. Adam is used with a learning rate of $10^{-3}$ for the main network and a reduced rate of $10^{-5}$ for feature extractor, setting the temperature parameter $\tau$ to $\frac{1}{100}$. For reproducibility, we fix random seeds in all experiments. We also assess time cost for both training and inference.

**Results.** In Table 3, BLCNet outperforms BBGM, Sinkhorn-NGMv2, and all baselines intended for bijective matching and achieves results on par with LinSATNet, with consid-

erably lower time. Our method's training and inference time are close to those of NGMv2 using Sinkhorn, yet with significantly better performance. Compared with LinSAT, it is slightly better while using far less compute. This gap amplifies when increasing iteration counts: LinSATNet (40 iters) gains only 0.0045% accuracy yet doubles overhead. We defer more details to Appendix C.1 for your reference.

### 3.2. Experiments on Portfolio Allocation

**Problem.** It aims to select the optimal distribution of assets in financial markets, by designing a portfolio that balances returns against risks. Following LinSATNet, we use StemGNN (Cao et al., 2020) as the backbone with two branches, one predicting future asset prices and the other the portfolio. BLCLayer is applied to the portfolio prediction branch to enforce constraints. The network is supervised by a weighted sum of maximizing the Sharpe ratio and minimizing the prediction error of prices.

**Constraint Formulation and Iterations in PA.** Each asset is assigned a non-negative weight representing the proportion of total investment allocated to that asset. The sum of these weights equal 1 (sum-to-one constraint). We introduce an expert preference constraint that requires the total allocation to a particular subset of assets, $\mathcal{C}$, to be at least a given threshold $p \in [0, 1]$. These constraints can be expressed as ($\mathbf{x}$ is the portfolio allocation): $\sum_{i=1}^{n} x_{(i)} = 1, \qquad \sum_{i\in\mathcal{C}} x_{(i)} \geq p$. The first constraint enforces the sum-to-one condition, while the second enforces the expert-driven preference for assets in set $\mathcal{C}$. Then we can build the BLCNet as discussed in Sec. 2.2 and GCLNet as in Sec. 2.3, with the same backbone among baselines.

**Experimental Details.** Given 120-day historical data, the network aims to build a portfolio with a maximized sharpe ratio for the next 120 days. The historical data includes real prices of 494 assets from the S&P 500 index from 2019-01-01 to 2020-12-30, while the model is tested on the real prices from 2021-03-01 to 2021-12-30. The subset of preferred stocks is set to C =(AAPL,MSFT, AMZN, TSLA, GOOGL, GOOG), and the preference ratio is set to p = 0.5.

For fairness, we strictly adhere to the configuration of LinSATNet, conducting 50 training epochs with a consistent batch size of 32 and a learning rate of $10^{-5}$ applied uniformly across all satisfiability layers. The training process employs a composite loss combining two components: 1) the mean squared error (MSE) for predicting future asset prices; and 2) the negative Sharpe ratio. To optimize performance while maintaining high inference speed, we implement constrained optimization with a tolerance threshold of $10^{-3}$, setting the temperature parameter $\tau$ to $\frac{1}{100}$ during training and reducing it to $\frac{1}{200}$ for inference phases. For computational efficiency analysis, we separately measure the execution time for both training and inference stages,

*Table 6.* Comprehensive experimental results. **Left:** Main benchmarks on Small, Medium, and Large scales. **Right:** Additional analyses including (Top) HardNet comparison, (Middle) Generalization, and (Bottom) Mixup.

**Main Benchmarks (Small / Medium / Large)**

| Scale | Setting (V, E, I) | Method | Obj. | Pct. | Time | T-Pct. |
|---|---|---|---|---|---|---|
| Small | (500, 400, 0) | Gurobi | -2919.76 | 1.000 | 0.192 | 1.000 |
| | | GLCNet | -2682.20 | 0.919 | 0.162 | 0.846 |
| | | GLCNet-UL | -2749.62 | 0.942 | 0.152 | 0.794 |
| | | GLCNet-SL | -2608.13 | 0.893 | 0.167 | 0.874 |
| | | SL-ADMM | **-2813.42** | **0.964** | 0.133 | 0.693 |
| | (500, 0, 400) | Gurobi | -11803.58 | 1.000 | 0.145 | 1.000 |
| | | GLCNet | -10900.49 | 0.923 | 0.159 | 1.094 |
| | | GLCNet-UL | -11168.67 | 0.946 | 0.165 | 1.136 |
| | | GLCNet-SL | -10651.20 | 0.902 | 0.154 | 1.058 |
| | | SL-ADMM | **-11366.26** | **0.963** | 0.177 | 1.215 |
| | (500, 200, 200) | Gurobi | -7506.81 | 1.000 | 0.168 | 1.000 |
| | | GLCNet | -6867.38 | 0.915 | 0.182 | 1.088 |
| | | GLCNet-UL | -7053.92 | 0.940 | 0.192 | 1.144 |
| | | GLCNet-SL | -6677.61 | 0.890 | 0.178 | 1.063 |
| | | SL-ADMM | **-7179.59** | **0.956** | 0.205 | 1.221 |
| Medium | (1k, 600, 0) | Gurobi | -14841.92 | 1.000 | 0.653 | 1.000 |
| | | GLCNet | -13421.70 | 0.904 | 0.214 | 0.327 |
| | | GLCNet-UL | -13821.09 | 0.931 | 0.206 | 0.316 |
| | | GLCNet-SL | -13190.91 | 0.889 | 0.222 | 0.339 |
| | | SL-ADMM | **-14156.82** | **0.954** | 0.186 | 0.285 |
| | (1k, 0, 600) | Gurobi | -26753.94 | 1.000 | 0.429 | 1.000 |
| | | GLCNet | -24144.09 | 0.902 | 0.177 | 0.413 |
| | | GLCNet-UL | -25032.86 | 0.936 | 0.190 | 0.441 |
| | | GLCNet-SL | -23653.69 | 0.884 | 0.158 | 0.367 |
| | | SL-ADMM | **-25649.54** | **0.959** | 0.188 | 0.439 |
| | (1k, 300, 300) | Gurobi | -20915.36 | 1.000 | 0.504 | 1.000 |
| | | GLCNet | -19002.86 | 0.909 | 0.224 | 0.445 |
| | | GLCNet-UL | -19580.54 | 0.936 | 0.221 | 0.438 |
| | | GLCNet-SL | -18643.32 | 0.891 | 0.213 | 0.422 |
| | | SL-ADMM | **-19862.06** | **0.950** | 0.180 | 0.456 |
| Large | (2k, 1600, 0) | Gurobi | -11699.66 | 1.000 | 4.492 | 1.000 |
| | | GLCNet | -10545.37 | 0.901 | 0.173 | 0.039 |
| | | GLCNet-UL | -10632.07 | 0.909 | 0.157 | 0.035 |
| | | GLCNet-SL | -10408.37 | 0.890 | 0.185 | 0.041 |
| | | SL-ADMM | **-10850.38** | **0.927** | 0.164 | 0.050 |
| | (2k, 0, 1600) | Gurobi | -45422.77 | 1.000 | 4.060 | 1.000 |
| | | GLCNet | -40885.95 | 0.900 | 0.167 | 0.041 |
| | | GLCNet-UL | -40538.46 | 0.892 | 0.150 | 0.037 |
| | | GLCNet-SL | -41327.91 | 0.910 | 0.176 | 0.043 |
| | | SL-ADMM | **-42237.73** | **0.930** | 0.151 | 0.043 |
| | (2k, 800, 800) | Gurobi | -28588.16 | 1.000 | 4.049 | 1.000 |
| | | GLCNet | -25995.50 | 0.909 | 0.160 | 0.040 |
| | | GLCNet-UL | -25831.12 | 0.904 | 0.149 | 0.037 |
| | | GLCNet-SL | -25384.57 | 0.888 | 0.172 | 0.043 |
| | | SL-ADMM | **-26437.47** | **0.925** | 0.145 | 0.046 |

**Comparison vs HardNet**

| Scale | Setting (V, E, I) | GLCNet Obj. | GLCNet Pct. | HardNet Obj. | HardNet Pct. |
|---|---|---|---|---|---|
| Small | (500, 400, 0) | -2813.42 | **0.964** | -2690.41 | 0.922 |
| | (500, 0, 400) | -11366.26 | **0.963** | -10808.18 | 0.916 |
| | (500, 200, 200) | -7179.59 | **0.956** | -6817.91 | 0.908 |
| Med. | (1k, 600, 0) | -14156.82 | **0.954** | -13321.22 | 0.898 |
| | (1k, 0, 600) | -25649.54 | **0.959** | -24409.23 | 0.912 |
| | (1k, 300, 300) | -19862.06 | **0.950** | -18721.75 | 0.895 |
| Large | (2k, 1600, 0) | -10850.38 | **0.927** | -10289.26 | 0.880 |
| | (2k, 0, 1600) | -42237.73 | **0.930** | -40138.74 | 0.884 |
| | (2k, 800, 800) | -26437.47 | **0.925** | -24829.96 | 0.869 |

**Cross-scale Generalization**

| Type | Setting (V, E, I) | Method | Obj. | Pct. | Time | T-Pct. |
|---|---|---|---|---|---|---|
| Gen. | (1k, 800, 0) | Gurobi | -5858.06 | 1.000 | 1.165 | 1.000 |
| | | GLCNet | -4690.51 | 0.811 | 0.165 | 0.142 |
| | | GLCNet-UL | -5076.31 | 0.866 | 0.171 | 0.146 |
| | | GLCNet-SL | -5111.89 | 0.871 | 0.161 | 0.138 |
| | | SL-ADMM | **-5601.01** | **0.956** | 0.172 | 0.148 |
| | (1k, 0, 800) | Gurobi | -23101.82 | 1.000 | 0.707 | 1.000 |
| | | GLCNet | -19909.72 | 0.885 | 0.171 | 0.241 |
| | | GLCNet-UL | -21568.04 | 0.951 | 0.161 | 0.228 |
| | | GLCNet-SL | -21877.13 | 0.957 | 0.166 | 0.235 |
| | | SL-ADMM | **-22274.77** | **0.964** | 0.115 | 0.163 |
| | (1k, 400, 400) | Gurobi | -13934.96 | 1.000 | 0.838 | 1.000 |
| | | GLCNet | -11808.58 | 0.871 | 0.188 | 0.224 |
| | | GLCNet-UL | -12739.52 | 0.922 | 0.193 | 0.230 |
| | | GLCNet-SL | -12906.78 | 0.935 | 0.194 | 0.231 |
| | | SL-ADMM | **-13261.90** | **0.952** | 0.099 | 0.119 |

**Supervised/Unsupervised Mixup**

| Type | Setting (V, E, I) | Method | Obj. | Pct. | Time | T-Pct. |
|---|---|---|---|---|---|---|
| Mixup | (1k, 800, 0) | Gurobi | -5858.06 | 1.000 | 1.165 | 1.000 |
| | | GLCNet | -4690.51 | 0.811 | 0.173 | 0.148 |
| | | GLCNet-UL | -5076.31 | 0.866 | 0.178 | 0.153 |
| | | GLCNet-SL | -5111.89 | 0.871 | 0.170 | 0.146 |
| | | SL-ADMM | **-5346.65** | **0.913** | 0.186 | 0.159 |
| | (1k, 0, 800) | Gurobi | -23101.82 | 1.000 | 0.707 | 1.000 |
| | | GLCNet | -19956.04 | 0.890 | 0.177 | 0.250 |
| | | GLCNet-UL | -21811.43 | 0.944 | 0.168 | 0.237 |
| | | GLCNet-SL | -21647.91 | 0.943 | 0.173 | 0.244 |
| | | SL-ADMM | **-21930.56** | **0.949** | 0.115 | 0.163 |
| | (1k, 400, 400) | Gurobi | -13934.96 | 1.000 | 0.838 | 1.000 |
| | | GLCNet | -11854.92 | 0.874 | 0.195 | 0.232 |
| | | GLCNet-UL | -12804.71 | 0.919 | 0.198 | 0.237 |
| | | GLCNet-SL | -12861.47 | 0.921 | 0.201 | 0.240 |
| | | SL-ADMM | **-12984.60** | **0.932** | 0.190 | 0.210 |

with training time encompassing full epoch iterations and inference time reflecting real-time prediction latency.

**Results.** In Table 4, BLCNet and GLCNet achieve SOTA with high efficiency. **1) Performance:** GLCNet attains the highest Sharpe ratio (3.053). Unlike LinSATNet, which degrades with fewer iterations, BLCNet and GLCNet with just 5 iterations outperform LinSATNet-100, demonstrating robust convergence properties even under extremely tight iteration budgets. **2) Efficiency:** BLCNet and GLCNet reduce training and inference time by approximately 20× and 30× compared to LinSATNet-100, respectively. **3)** Constrained methods significantly outperform the unconstrained baseline. See Appendix B.4 for details.

## 3.3. Experiments on Linear Programming

**Problem Formulation.** Linear programming (LP) serves as a canonical framework for optimization:

$$\min_{\mathbf{x} \geq \mathbf{0}} \mathbf{c}^\top \mathbf{x} \quad \text{s.t.} \quad \mathbf{Wx} = \mathbf{b}. \tag{19}$$

Traditional LP solvers (e.g., Simplex) guarantee optimal solutions but often suffer from exponential time complexity in worst-case scenarios. While recent works have explored neural approximations for LP (Amos & Kolter, 2017), a fully end-to-end neural solver remains underexplored. In this section, we present GLCNet, employing a GNN backbone, as an efficient end-to-end solver for LP instances.

**Experimental Setup.** To evaluate performance across varying complexities, we generate synthetic LP instances with problem scales ranging from 100 to 2,000 variables. We ensure feasibility via a constructive approach: random feasible solutions are sampled first, from which equality constraints are derived. Inequality constraints are then generated with random offsets to maintain feasibility bounds. We formulate the objective functions by sampling from a normal distribution and filter out invalid cases (e.g., unbounded problems). We adopt four metrics for evaluation: 1) **Objective Score:** The raw optimization value; 2) **Optimality Gap:** The ratio of our attained score to Gurobi's reference optimal; 3) **Time Cost:** Wall-clock execution time; and 4) **Speedup Ratio :** Our runtime normalized against Gurobi's.

*Architecture Details.* GLCNet employs a bipartite GNN backbone with parallel 1D convolutional encoders to process heterogeneous constraints. These features are fused with objective weights via normalized matrix multiplication to yield variable-aligned embeddings, accommodating varying constraint configurations (see Appendix D).

**Baselines and Variants.** We benchmark against Gurobi and HardNet, a SOTA framework ensuring hard constraint satisfaction via a differentiable enforcement layer. For our proposed GLCNet, we evaluate four variants: GLCNet (Random) uses an untrained backbone where constraints are met solely via projection; GLCNet-UL is trained via unsupervised objective maximization; GLCNet-SL is trained via supervised MSE minimization against Gurobi solutions; and GLCNet-SL-ADMM refines the supervised predictions using a lightweight ADMM post-process.

**Main Results.** Table 6 presents a comprehensive evaluation. We analyze the performance from three perspectives:

*1) Efficiency vs. Traditional Solvers (Table 6 Left).* While GLCNet exhibits slower performance than Gurobi on small-scale instances, it demonstrates significant efficiency advantages as the problem scale increases. Gurobi's solving time grows markedly with scale; in contrast, GLCNet maintains stable efficiency. On large-scale problems, GLCNet delivers **5× to 20× acceleration** compared to the CPU-based Gurobi solver while achieving a 0.95 success rate and competitive objective scores. In practical deployments, Our methods enjoy the capability of test-time handling a batch of LP instances simultaneously i.e., instance batch level parallelization, though for fairness we currently set the batch size by 1. This feature can further significantly reduce the mean solving time by enlarging the batch size, which is an off-the-shelf advantage for modern GPU clusters.

*2) Comparison with Hard-Constrained Baselines (Table 6 Right-Top).* We explicitly compare GLCNet against **Hard-Net** (Min & Azizan, 2024), a recent SOTA method for enforcing hard constraints. Across all scales and constraint types, GLCNet consistently achieves higher **Objective Pct.**. Furthermore, regarding computational cost, GLCNet significantly outperforms HardNet. As detailed in Appendix **??**, HardNet relies on generic solver interfaces for projection, leading to slower training (approx. 15× slower per epoch). In contrast, GLCNet exploits the fixed structure of constraint matrices **W** to precompute factorizations, enabling highly efficient, GPU-accelerated projections.

**Further Studies.** We conduct ablation studies to explore the generalization and learning efficiency of our GNN backbone (Table 6 Right-Middle & Bottom).

*1) Cross-Scale Generalization & Test-Time Scale-up (TTS).* A key challenge in neural combinatorial optimization is generalizing to problem scales larger than those seen during training. Table 6 (Right-Middle) reports the results of models trained on small instances but tested on larger ones. We introduce TTS, which enables unsupervised fine-tuning via backpropagation during inference. Results show that direct inference (GLCNet) may suffer a performance drop on out-of-distribution scales. However, enabling TTS effectively bridges this gap, significantly improving the objective score. This demonstrates that our model can adaptively refine its solutions at test time, offering a flexible trade-off between inference speed and solution quality.

*2) Supervised/Unsupervised Mixup Strategy.* Obtaining ground-truth labels for large-scale LP instances is computationally prohibitive. To address this, we evaluate a "Mixup" training strategy: unsupervised pre-training followed by supervised fine-tuning. As shown in Table 6 (Right-Bottom), the Mixup strategy performs comparably to fully supervised training but with significantly reduced dependency on labeled data. This suggests that unsupervised pre-training effectively learns the geometric structure of the feasible region, serving as a robust warm start for the final solver. This is particularly valuable for industrial applications where "optimal labels" are scarce or expensive to generate.

## 4. Conclusion and Outlook

We have presented a framework for integrating linear constraints for neural networks, by developing differentiable layers. The experiments on general LP with linear constraints, and special combinatorial optimization tasks showcase its cost-effectiveness. Our approach can be beyond standard LP, as it only requires the constraints to be linear rather than the objective, which we leave for future work.

## Acknowledgment

The work was in part supported by NSFC 92370201 and Scientific Research Innovation Capability Support Project of China Ministry of Education SRICSPYF-ZY2025019.

## Impact Statement

This work bridges the gap between deep learning and constrained optimization by introducing a general, efficient framework for enforcing linear constraints in neural networks. By ensuring model outputs strictly adhere to mandatory requirements, our approach facilitates the deployment of AI in safety-critical and high-stakes domains such as finance, supply chain management, and energy distribution, where constraint violations can lead to system failures. Furthermore, our method achieves significant speedups over traditional solvers on large-scale problems, enabling real-time decision-making and contributing to energy-efficient computing. However, while this approximation capability offers substantial economic and operational benefits, practitioners must remain vigilant regarding the trade-off between inference speed and global optimality, and ensure that the encoded constraints do not perpetuate historical biases in automated decision-making.

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

# Appendix

## A. Related Works

We first discuss works on handling constraints for neural networks, especially for constrained problems. We also introduce the involved techniques i.e., optimal transport and Bregman projection.

**Differentiable Optimization Methods.** These solvers aim to integrate traditional optimization methods with neural networks by utilizing backpropagation to compute gradients. Notably, (Amos & Kolter, 2017) demonstrated the differentiability at optimal solutions by applying KKT conditions, particularly for quadratic programming. Concurrently, (Anh-Nguyen et al., 2024) proposed learning generalized linear programming value functions (GVFs) to approximate LP solutions, highlighting the role of implicit optimization in constraint satisfaction. Building on this foundation, further advancement (Pogančić et al., 2020; Berthet et al., 2020) focused on gradient approximation techniques. These methods allow combinatorial solvers to estimate gradients using forward passes or random perturbations, broadening the scope of differentiable optimization.

**Heuristic Search.** These methods focus on generating feasible solutions through search strategies. Two primary strategies have emerged: **Autoregressive search methods**, exemplified by models like (Bello et al., 2017) enforce hard constraints by sequentially constructing solutions, though their sequential nature introduces scalability bottlenecks. Meanwhile, **non-autoregressive search approaches** address scalability by assuming conditional independence among variables. Works like DIFUSCO (Sun & Yang, 2023) leverage heatmap-based search with iterative denoising for parallel decoding, but face inherent trade-offs between solution quality and multimodality (Fu et al., 2021; Joshi et al., 2019; Geisler et al., 2022).

**Reinforcement Learning and Generative methods.** Reinforcement learning (RL) represents one prominent line of research, especially applied to combinatorial optimization problems ranging from the traveling salesman problem (Khalil et al., 2017), vehicle routing (Nazari et al., 2018) to job scheduling (Chen & Tian, 2019). While RL shows remarkable adaptability, its performance hinges on carefully engineered reward functions. A parallel research direction explores **diffusion models**, originally designed for continuous data generation (Sohl-Dickstein et al., 2015), now adapted to discrete domains through binomial or multinomial perturbations (Austin et al., 2021). These discrete diffusion frameworks have shown promise in constraint satisfaction tasks, particularly through works like (Niu et al., 2020), which developed permutation-invariant score-based models for graph-based optimization. Building on these foundations, recent efforts combine diffusion with graph neural networks (Sun & Yang, 2023), though computational complexity remains a barrier for NP-hard problems.

## B. Constrained Layer Design via Divergence based Implicit Optimization

### B.1. Derivation of BLCNet via KL Divergence-Based Implicit Optimization

To find the probability distribution $\mathbf{x}$ that minimizes the Kullback-Leibler (KL) divergence subject to the constraint $\sum_i x_{(i)} = 1$, we can use the method of Lagrange multipliers. The optimization problem is formally stated as:

$$\min_{\mathbf{x}} \mathrm{KL}(\mathbf{x}|\mathbf{y}) = \sum_{i=1}^{n} \left( x_{(i)} \log \frac{x_{(i)}}{k_i} - x_{(i)} + k_i \right) \quad \text{s.t.} \sum_{i=1}^{n} x_{(i)} = 1$$

Introduce the Lagrange multiplier $\lambda$ and construct the Lagrangian function:

$$L(\mathbf{x}, \lambda) = \sum_{i=1}^{n} (x_{(i)} \log \left( \frac{x_{(i)}}{k_i} \right) - x_{(i)} + k_i) + \lambda \left( \sum_{i=1}^{n} x_{(i)} - 1 \right)$$

Take the partial derivative of $L(\mathbf{x}, \lambda)$ with respect to each $x_{(i)}$:

$$\frac{\partial L}{\partial x_{(i)}} = \log \left( \frac{x_{(i)}}{k_i} \right) + \lambda = 0$$

Solving for $x_{(i)}$, we get:

$$\log \left( \frac{x_{(i)}}{k_i} \right) = -\lambda$$

Taking the exponential of both sides:

$$\frac{x_{(i)}}{k_i} = e^{-\lambda} \implies x_{(i)} = k_i e^{-\lambda}$$

Substitute $x_{(i)}$ into the constraint $\sum_{i=1}^{n} x_{(i)} = 1$:

$$\sum_{i=1}^{n} x_{(i)} = \sum_{i=1}^{n} k_i e^{-\lambda} = e^{-\lambda} \sum_{i=1}^{n} k_i = 1$$

Solving for $e^{-\lambda}$, we get:

$$e^{-\lambda} = \frac{1}{\sum_{i=1}^{n} k_i}$$

Thus:

$$x_{(i)} = k_i \cdot \frac{1}{\sum_{j=1}^{n} k_j}$$

The optimal solution is given by:

$$x_{(i)} = \frac{k_i}{\sum_{j=1}^{n} k_j}$$

When $k_i$ takes the exponential form $k_i = e^{y_i/\tau}$ (where $\tau > 0$ is a constant), the optimal solution becomes:

$$x_{(i)} = \frac{e^{y_i/\tau}}{\sum_{j=1}^{n} e^{y_j/\tau}},$$

which corresponds to the **softmax function** scaled by the parameter $\tau$. Here, $\tau$ controls the sharpness of the distribution.

### B.2. Derivation of GLCNet via Euclidean Distance-Based Implicit Optimization

To solve the constrained optimization problem

$$\min_{\mathbf{x}} \frac{1}{2} \|\mathbf{x} - \mathbf{y}\|^2 \quad \text{s.t.} \mathbf{W}\mathbf{x} = \mathbf{b},$$

we employ the method of Lagrange multipliers. Construct the Lagrangian function with multiplier vector $\boldsymbol{\lambda}$:

$$\mathcal{L}(\mathbf{x}, \boldsymbol{\lambda}) = \frac{1}{2} \|\mathbf{x} - \mathbf{y}\|^2 + \boldsymbol{\lambda}^\top (\mathbf{W}\mathbf{x} - \mathbf{b})$$

Taking the gradient with respect to $\mathbf{x}$ and setting it to zero yields:

$$\nabla_{\mathbf{x}} \mathcal{L} = (\mathbf{x} - \mathbf{y}) + \mathbf{W}^\top \boldsymbol{\lambda} = 0$$

Solving for $\mathbf{x}$ gives the intermediate solution:

$$\mathbf{x} = \mathbf{y} - \mathbf{W}^\top \boldsymbol{\lambda}$$

Substituting this expression into the constraint $\mathbf{W}\mathbf{x} = \mathbf{b}$ produces:

$$\mathbf{W}\mathbf{y} - \mathbf{W}\mathbf{W}^\top \boldsymbol{\lambda} = \mathbf{b}$$

Rearranging terms to solve for the Lagrange multipliers, we obtain:

$$\mathbf{W}\mathbf{W}^\top \boldsymbol{\lambda} = \mathbf{W}\mathbf{y} - \mathbf{b}$$

Under the assumption that $\mathbf{W}\mathbf{W}^\top$ is invertible, the multipliers become:

$$\boldsymbol{\lambda} = (\mathbf{W}\mathbf{W}^\top)^{-1}(\mathbf{W}\mathbf{y} - \mathbf{b})$$

Substituting back into the expression for $\mathbf{x}$ yields the optimal solution:

$$\mathbf{x}^* = \mathbf{y} - \mathbf{W}^\top(\mathbf{W}\mathbf{W}^\top)^{-1}(\mathbf{W}\mathbf{y} - \mathbf{b})$$

To verify constraint satisfaction, we compute:

$$\mathbf{W}\mathbf{x}^* = \mathbf{W}\mathbf{y} - \mathbf{W}\mathbf{W}^\top(\mathbf{W}\mathbf{W}^\top)^{-1}(\mathbf{W}\mathbf{y} - \mathbf{b}) = \mathbf{b}$$

This confirms that the solution strictly satisfies the equality constraint $\mathbf{W}\mathbf{x} = \mathbf{b}$.

### B.3. Derivation of Sigmoid function via KL Divergence Minimization

Consider the optimization problem:

$$\min_x \mathrm{KL}\left([x, 1 - x] \,\|\, \left[e^{y/2}, e^{-y/2}\right]\right) \quad \text{s.t.} \quad x \in (0, 1)$$

where the KL divergence is defined as:

$$\mathrm{KL}(\mathbf{X} \,\|\, \mathbf{Q}) = \sum_i \left( x_{(i)} \log \frac{x_{(i)}}{\mathbf{Q}_{(i)}} - x_{(i)} + \mathbf{Q}_{(i)} \right).$$

Substituting $\mathbf{P} = [x, 1 - x]$ and $\mathbf{Q} = \left[e^{y/2}, e^{-y/2}\right]$, the divergence expands to:

$$\mathrm{KL}(\mathbf{P} \,\|\, \mathbf{Q}) = x \log \frac{x}{e^{y/2}} - x + e^{y/2} + (1 - x) \log \frac{1 - x}{e^{-y/2}} - (1 - x) + e^{-y/2}.$$

Simplifying terms and removing constants independent of $x$, the objective reduces to:

$$f(x) = x \log x + (1 - x) \log(1 - x) - yx.$$

To find the minimum, take the derivative of $f(x)$ with respect to $x$:

$$\frac{df}{dx} = \log x - \log(1 - x) - y = \log \frac{x}{1 - x} - y.$$

Setting the derivative to zero gives the critical condition:

$$\log \frac{x}{1 - x} = y \implies \frac{x}{1 - x} = e^y.$$

Solving for $x$, we directly obtain the Sigmoid function:

$$x = \frac{e^y}{1 + e^y} = \frac{1}{1 + e^{-y}} = \sigma(y).$$

### B.4. Ablation Study for Portfolio Allocation

We also give the ablation test in Table 8.

This demonstrates that the Sigmoid function $\sigma(y)$ minimizes the KL divergence when $Q = \left[e^{y/2}, e^{-y/2}\right]$.

*Table 7.* F1, Train and Inference Time of Graph Matching on Pascal VOC Keypoint (Unfiltered).

| GM Network | Constraint Technique | Max Iter | $\tau$ | F1 | Train | Inference |
|---|---|---|---|---|---|---|
| BBGM | (Pogančić et al., 2020) | - | - | 0.5516 | 1h 12m | 1h 21m |
| NGMv2 | Sinkhorn | 10 | 0.01 | 0.5453 | 1h 25m | 1h 34m |
| NGMv2 | LinSAT | 5 | 0.01 | 0.5858 | 1h 56m | 2h 15m |
| NGMv2 | LinSAT | 10 | 0.01 | 0.5839 | 2h 19m | 2h 37m |
| NGMv2 | LinSAT | 20 | 0.01 | 0.5830 | 3h 14m | 3h 27m |
| NGMv2 | LinSAT | 40 | 0.01 | 0.5894 | 4h 52m | 4h 58m |
| NGMv2 | LinSAT | 10 | 0.02 | 0.6007 | 2h 25m | 2h 42m |
| NGMv2 | LinSAT | 10 | 0.1 | 0.5041 | 2h 21m | 2h 37m |
| NGMv2 | BLCNet | 5 | 0.01 | 0.5972 | 1h 28m | 1h 36m |
| NGMv2 | BLCNet | 10 | 0.01 | 0.6012 | 1h 37m | 1h 47m |
| NGMv2 | BLCNet | 20 | 0.01 | 0.5937 | 1h 46m | 1h 51m |
| NGMv2 | BLCNet | 10 | 0.02 | 0.5920 | 1h 36m | 1h 41m |
| NGMv2 | BLCNet | 10 | 0.1 | 0.5921 | 1h 38m | 1h 47m |

*Table 8.* Sharpe Ratio, Train, and Inference Time (in Seconds) for the Portfolio Allocation Methods.

| Layer | Train Max-Iter | Test Max-Iter | $\tau_{train}$ | $\tau_{test}$ | Sharpe Ratio | Train Time (sec) | Inference Time (sec) |
|---|---|---|---|---|---|---|---|
| Unconstrained | - | - | - | - | 2.085 | 138.02s | 7.77s |
| LinSAT | 5 | 5 | 1/10 | 1/200 | 2.124 | 595.98s | 44.45s |
| LinSAT | 5 | 5 | 1/10 | 1/100 | 1.646 | 620.92s | 45.80s |
| LinSAT | 5 | 5 | 1/10 | 1/50 | 1.426 | 603.75s | 43.81s |
| LinSAT | 5 | 5 | 1/10 | 1/20 | 1.335 | 581.46s | 42.99s |
| LinSAT | 5 | 5 | 1/10 | 1/10 | 1.311 | 561.24 | 41.65 |
| LinSAT | 5 | 5 | 1/100 | 1/100 | 1.891 | 571.58s | 40.62s |
| LinSAT | 5 | 5 | 1/100 | 1/200 | 2.392 | 631.66s | 47.69s |
| LinSAT | 10 | 10 | 1/100 | 1/200 | 2.560 | 977.35s | 69.02s |
| LinSAT | 100 | 100 | 1/100 | 1/200 | 2.752 | 4450.98s | 447.43s |
| BLCNet | 1 | 1 | 1/100 | 1/200 | 2.808 | 170.37s | 10.44s |
| BLCNet | 5 | 5 | 1/10 | 1/200 | 2.660 | 229.58s | 13.78s |
| BLCNet | 5 | 5 | 1/10 | 1/100 | 2.381 | 229.87s | 13.91s |
| BLCNet | 5 | 5 | 1/10 | 1/50 | 2.249 | 229.58s | 13.92s |
| BLCNet | 5 | 5 | 1/10 | 1/20 | 2.191 | 233.60s | 14.02s |
| BLCNet | 5 | 5 | 1/10 | 1/10 | 2.175 | 198.40s | 11.77s |
| BLCNet | 5 | 5 | 1/100 | 1/200 | 2.758 | 224.48s | 17.20s |
| BLCNet | 1 | 10 | 1/100 | 1/200 | 2.780 | 150.70s | 17.43s |
| BLCNet | 10 | 1 | 1/100 | 1/200 | 2.773 | 274.07s | 8.52s |
| BLCNet | 10 | 10 | 1/100 | 1/200 | 2.755 | 303.57s | 19.44s |
| GLCNet | 5 | 5 | - | - | 3.000 | 259.60s | 14.52s |
| GLCNet | 10 | 10 | - | - | 3.027 | 403.67s | 27.38s |

## B.5. Derivation of $\tanh(y)$ via KL Divergence Minimization

To solve the optimization problem of minimizing the KL divergence given $y$, we start by defining the probability distributions:

$$\mathbf{X} = \left[ \frac{x+1}{2}, \frac{1-x}{2} \right], \quad \mathbf{Q} = \left[ e^y, e^{-y} \right]$$

The optimization target is defined as:

$$\min_x \text{KL}(\mathbf{X} \parallel \mathbf{Q}) \qquad x_{(i)} \geq 0$$

The KL divergence is defined as:

$$\text{KL}(\mathbf{X} \parallel \mathbf{Q}) = \sum_{i=1}^{2} \left[ x_{(i)} \log \frac{x_{(i)}}{\mathbf{Q}_i} - x_{(i)} + \mathbf{Q}_i \right]$$

Substituting the components of $P$ and $Q$, we expand the objective function:

$$\text{KL}(P \parallel Q) = \frac{x+1}{2} \log \left( \frac{x+1}{2} \right) - \frac{x+1}{2} \log e^y - \frac{x+1}{2} + e^y$$
$$+ \frac{1-x}{2} \log \left( \frac{1-x}{2} \right) - \frac{1-x}{2} \log e^{-y} - \frac{1-x}{2} + e^{-y}$$

Simplifying using $\log e^y = y$ and combining terms:

$$\text{KL}(P \parallel Q) = \frac{x+1}{2} \log \left( \frac{x+1}{2} \right) + \frac{1-x}{2} \log \left( \frac{1-x}{2} \right) - \frac{(x+1)y}{2} + \frac{(1-x)y}{2} - 1 + e^y + e^{-y}$$
$$= \left[ \frac{x+1}{2} \log \left( \frac{x+1}{2} \right) + \frac{1-x}{2} \log \left( \frac{1-x}{2} \right) \right] - xy - 1 + e^y + e^{-y}$$

Taking the derivative with respect to $x$ and setting it to zero:

$$\frac{d}{dx} \text{KL} = \frac{1}{2} \log \left( \frac{x+1}{1-x} \right) - y = 0$$

Solving this equation:

$$\log \left( \frac{x+1}{1-x} \right) = 2y$$
$$\frac{x+1}{1-x} = e^{2y}$$
$$x = \frac{e^{2y}-1}{e^{2y}+1} = \tanh(y)$$

Therefore, the optimal solution is:

$$x = \tanh(y)$$

### B.6. Proof of Eq. 9

Now, given that $\mathbf{w}_{(l)} \in \{0,1\}^n$, we aim to solve the following optimization problem:

$$\min_{\mathbf{x} \geq 0} \text{KL}(\mathbf{x} \mid \mathbf{x}^{(l)}) \quad \text{s.t.} \quad \mathbf{w}_{(l)}^\top \mathbf{x} = b_{(l)}$$

We introduce a Lagrange multiplier $\lambda_l$ for the constraint $\mathbf{w}_{(l)}^\top \mathbf{x} = b_{(l)}$. The Lagrangian for this problem is:

$$\mathcal{L}(\mathbf{x}, \lambda_l) = \sum_{i=1}^{n} \left( x_{(i)} \log \frac{x_{(i)}}{x_{(i)}^{(l)}} - x_{(i)} + x_{(i)}^{(l)} \right) + \lambda_l (\mathbf{w}_{(l)}^\top \mathbf{x} - b_{(l)})$$

The gradient of the Lagrangian with respect to $\mathbf{x}$ is:

$$\frac{\partial \mathcal{L}}{\partial x_{(i)}} = \log \frac{x_{(i)}}{x_{(i)}^{(l)}} + \lambda_l w_{li} = 0$$

This simplifies to:

$$x_{(i)} = x_{(i)}^{(l)} e^{-\lambda_l w_{li}}$$

- If $w_{li} = 0$, we have:

$$x_{(i)} = x_{(i)}^{(l)} = (1 - w_{li}) \cdot x_{(i)}^{(l)}$$

This means that when $w_{li} = 0$, the component $x_{(i)}$ is not updated and remains equal to its initial value $x_{(i)}^{(l)}$.

- If $w_{li} = 1$, we get:

$$x_{(i)} = x_{(i)}^{(l)} e^{-\lambda_l}$$

This means that when $w_{li} = 1$, the component $x_{(i)}$ is updated according to the exponential factor $e^{-\lambda_l}$.

Since we know that $\mathbf{w}_{(l)}^\top \mathbf{x} = b_{(l)}$, we can write:

$$\mathbf{w}_{(l)}^\top \mathbf{x} = \sum_i x_{(i)} = e^{-\lambda_l} \left( \sum_i x_{(i)}^{(l)} \right) = b_{(l)}$$

Thus:

$$e^{-\lambda_l} = \frac{b_{(l)}}{\sum_i x_{(i)}^{(l)}} = \frac{b_{(l)}}{\mathbf{w}_{(l)}^\top \mathbf{x}^{(l)}}$$

Therefore, we have:

$$x_{(i)} = x_{(i)}^{(l)} \cdot \frac{b_{(l)}}{\mathbf{w}_{(l)}^\top \mathbf{x}^{(l)}} = w_{li} x_{(i)}^{(l)} \cdot \frac{b_{(l)}}{\mathbf{w}_{(l)}^\top \mathbf{x}^{(l)}}$$

Combining the cases for $w_{li} = 0$ and $w_{li} = 1$, we arrive at the final solution:

$$\mathbf{x}^{(l+1)} = (\mathbf{1}_n - \mathbf{w}_{(l)}) \odot \mathbf{x}^{(l)} + \mathbf{w}_{(l)} \odot \frac{b_{(l)}}{\mathbf{w}_{(l)}^\top \mathbf{x}^{(l)}} \mathbf{x}^{(l)}$$

In this expression, $\mathbf{w}_{(l)}$ is a binary vector that determines which components of $\mathbf{x}$ are updated. For components where $w_{li} = 0$, the corresponding $x_{(i)}$ remains unchanged, while for components where $w_{li} = 1$, the update is computed as $x_{(i)} = x_{(i)}^{(l)} \cdot \frac{b_{(l)}}{\mathbf{w}_{(l)}^\top \mathbf{x}^{(l)}}$.

## C. Ablation studies

### C.1. Ablation Study for Graph Matching

We give the ablation results in Table 7.

### C.2. Profiling the Cost of Matrix Inversion in BLCLayer

One potential concern for BLCLayer is the cost of computing matrix inverses involving $\mathbf{W}\mathbf{W}^\top$. Howerver, in our applications, the rank and dimension of $\mathbf{W}$ are very small, and the inversion step is negligible in practice compared to the rest of the network computation.

*Table 9.* Profiling the cost of matrix inversion on the partial graph matching and portfolio allocation tasks. "Proj." denotes the total projection time per forward pass, and "Inv." the time spent in the matrix inversion / linear solve.

| Task | Avg. proj. time (ms) | Avg. inv. time (ms) | Inv. ratio (%) |
|---|---|---|---|
| Graph matching | 148.5520 | 0.0003 | 0.0002 |
| Portfolio | 2.6887 | 0.0001 | 0.0001 |

As an illustration, Table 9 reports the profiling results on the partial graph matching and portfolio allocation tasks. For each forward pass, we measure (i) the total time spent in the BLCLayer projection and (ii) the time spent specifically on the $\mathbf{W}\mathbf{W}^\top$ inversion / linear solve, and average these over many batches.

In both cases the inversion accounts for a vanishingly small fraction of the overall projection time. For our LP experiments, the constraint matrices are larger but still structured; we exploit this by precomputing a factorization of $\mathbf{W}\mathbf{W}^\top$ once and reusing it across all training and inference steps. Compared to classical LP solvers that repeatedly solve large linear systems (e.g., in simplex or interior-point iterations), GLCLayer only needs this preprocessing step once, and can amortize the cost over many problem instances and forward passes.

When scaling to larger problems, two primary factors influence resource consumption:

- **Constraint Sparsity.** Real-world constraint matrices $\mathbf{W}$ are often highly sparse. Leveraging sparse tensor formats and graph-based encoders significantly reduces storage and computation. Empirically, we find that problem dimensions up to $\mathcal{O}(10^4)$ remain feasible within a single training pipeline.

- **High-dimensional Outputs.** For extremely high-dimensional output vectors $\mathbf{x}$, memory bottlenecks can be mitigated via block-wise autoregressive strategies. By partitioning $\mathbf{x}$ and enforcing constraints locally, the model can scale to massive industrial tasks.

Consequently, our layer benefits directly from sparse linear algebra, multi-GPU training, and sequential prediction architectures.

## D. Backbone for GLCNet

The backbone for GLCNet is shown in 1.

*Listing 1.* **Backbone for GLCNet**

```python
class BipartiteGraphConvolution(nn.Module):
    """
    Partial bipartite graph convolution (either left-to-right or right-to-left).
    """
    def __init__(self, emb_size, activation, initializer,
                 right_to_left=False, gather_first=False):
        super().__init__()

        self.emb_size = emb_size
        self.activation = activation
        self.right_to_left = right_to_left
        self.gather_first = gather_first

        # feature layers
        self.feature_module = nn.Sequential(
            nn.Linear(in_features=emb_size, out_features=self.emb_size),
            self.activation,
            nn.Linear(in_features=self.emb_size, out_features=self.emb_size),
        )
        # output_layers
        self.output_module = nn.Sequential(
            nn.Linear(in_features=emb_size * 2, out_features=self.emb_size),
            self.activation,
```

```python
24                 nn.Linear(in_features=self.emb_size, out_features=self.emb_size),
25             )
26
27             # Apply initializer
28             for module in [self.feature_module, self.output_module]:
29                 for layer in module:
30                     if isinstance(layer, nn.Linear):
31                         initializer(layer.weight)
32                         if layer.bias is not None:
33                             nn.init.zeros_(layer.bias)
34
35     def forward(self, inputs):
36         left_features, edge_indices, edge_features, right_features = inputs
37
38         if self.right_to_left:
39             scatter_dim, gather_dim = 0, 1
40             prev_features, gather_feautures = left_features, right_features
41         else:
42             scatter_dim, gather_dim = 1, 0
43             prev_features, gather_feautures = right_features, left_features
44
45         # compute joint features
46         if self.gather_first:
47             gathered = torch.index_select(gather_feautures, dim=0,
    index=edge_indices[gather_dim])
48             joint_features = self.feature_module(torch.cat([edge_features, gathered],
    dim=1))
49         else:
50             gather_features = self.feature_module(gather_feautures)
51             joint_features = edge_features * torch.index_select(
52                 gather_features, dim=0, index=edge_indices[gather_dim]
53             )
54
55         # perform convolution
56         conv_output = torch.zeros(
57             prev_features.shape[0], self.emb_size, device=prev_features.device
58         )
59         conv_output.index_add_(0, edge_indices[scatter_dim], joint_features)
60
61         # output layer
62         output = self.output_module(torch.cat([conv_output, prev_features], dim=1))
63
64         return output
65
66
67 class ExtendedBipartiteGraphConvolution(nn.Module):
68     def __init__(self, emb_size, activation, initializer, right_to_left=False,
    gather_first=False):
69         super().__init__()
70
71         self.emb_size = emb_size
72         self.activation = activation
73         self.right_to_left = right_to_left
74         self.gather_first = gather_first
75
76         # feature layers
77         self.feature_module = nn.Sequential(
78             nn.Linear(in_features=emb_size, out_features=self.emb_size),
79             self.activation,
80             nn.Linear(in_features=self.emb_size, out_features=self.emb_size),
81         )
82         self.feature_module_self = nn.Sequential(
83             nn.Linear(in_features=emb_size, out_features=self.emb_size, bias=False),
84             self.activation,
85             nn.Linear(in_features=self.emb_size, out_features=self.emb_size),
```

```
86              )
87
88          # output_layers
89          self.output_module = nn.Sequential(
90              nn.Linear(in_features=emb_size * 2 + emb_size, out_features=self.emb_size),
91              self.activation,
92              nn.Linear(in_features=self.emb_size, out_features=self.emb_size),
93          )
94
95          # Apply initializer
96          for module in [self.feature_module, self.feature_module_self,
     self.output_module]:
97              for layer in module:
98                  if isinstance(layer, nn.Linear):
99                      initializer(layer.weight)
100                     if layer.bias is not None:
101                         nn.init.zeros_(layer.bias)
102
103      def forward(self, inputs):
104          left_features, edge_indices, edge_features, right_features, self_edges = inputs
105
106          if self.right_to_left:
107              scatter_dim, gather_dim = 0, 1
108              prev_features, gather_feautures = left_features, right_features
109          else:
110              scatter_dim, gather_dim = 1, 0
111              prev_features, gather_feautures = right_features, left_features
112
113          # compute joint features
114          if self.gather_first:
115              gathered = torch.index_select(gather_feautures, dim=0,
     index=edge_indices[gather_dim])
116              joint_features = self.feature_module(torch.cat([edge_features, gathered],
     dim=1))
117          else:
118              gather_features = self.feature_module(gather_feautures)
119              joint_features = edge_features * torch.index_select(
120                  gather_features, dim=0, index=edge_indices[gather_dim]
121              )
122
123          # perform convolution
124          conv_output = torch.zeros(
125              prev_features.shape[0], self.emb_size, device=prev_features.device
126          )
127          conv_output.index_add_(0, edge_indices[scatter_dim], joint_features)
128
129          # self convolution
130          self_out = None
131          if isinstance(self_edges, torch.Tensor): # Q_matrix
132              if self.gather_first:
133                  raise ValueError("Q in matrix is not supported for mixed-integer
     extension.")
134              self_out = self.feature_module_self(prev_features)
135              self_out = torch.matmul(self_edges, self_out)
136          elif isinstance(self_edges, tuple) and len(self_edges) == 2: # self_inds,
     self_feats
137              self_inds, self_feats = self_edges
138              if self.gather_first:
139                  self_gathered = torch.index_select(prev_features, dim=0,
     index=self_inds[1])
140                  self_gathered = self.feature_module_self(torch.cat([self_feats,
     self_gathered], dim=1))
141              else:
142                  self_gathered = self_feats * torch.index_select(
143                      self.feature_module_self(prev_features),
```

```
144                        dim=0,
145                        index=self_inds[1]
146                    )
147                self_out = torch.zeros(
148                    prev_features.shape[0], self_gathered.shape[1],
         device=prev_features.device
149                )
150                self_out.index_add_(0, self_inds[0], self_gathered)
151
152            # output layer
153            to_concat = [prev_features, conv_output]
154            if self_edges is not None:
155                to_concat += [self_out]
156            output = self.output_module(torch.cat(to_concat, dim=1))
157
158            return output
```

## E. Convergence Proof

**Theorem E.1** (Proposition). *The sequence $\{\mathbf{x}^{(k)}\}$ generated by the Bregman iterative projection method converges to the unique solution $\mathbf{x}^*$ of Eq. 7 provided that the feasible set $\mathcal{C}$ is nonempty.*

*Proof.* The KL-divergence satisfies a fundamental identity known as the three-point property:

$$\mathrm{KL}(\mathbf{x} \mid \mathbf{z}) = \mathrm{KL}(\mathbf{x} \mid \mathbf{y}) + \mathrm{KL}(\mathbf{y} \mid \mathbf{z}) + \langle \nabla f(\mathbf{y}) - \nabla f(\mathbf{z}), \mathbf{x} - \mathbf{y} \rangle,$$

where $f(\mathbf{x}) = \sum_i (x_{(i)} \log x_{(i)} - x_{(i)})$ is the convex function. This identity originates from the more general Bregman divergence(Bregman, 1967), of which the KL-divergence is a special case.

Let $\mathbf{x}^*$ be the unique optimal solution (uniqueness guaranteed by closed and convex set $\mathcal{C}$). Applying the identity with $\mathbf{x} = \mathbf{x}^*$, $\mathbf{y} = \mathbf{x}^{(k+1)}$, $\mathbf{z} = \mathbf{x}^{(k)}$, and noting that $\mathbf{x}^* \in \mathcal{C} \subset \mathcal{C}_l$, we obtain:

$$\mathrm{KL}(\mathbf{x}^* \mid \mathbf{x}^{(k)}) = \mathrm{KL}(\mathbf{x}^* \mid \mathbf{x}^{(k+1)}) + \mathrm{KL}(\mathbf{x}^{(k+1)} \mid \mathbf{x}^{(k)}) + \langle \nabla f(\mathbf{x}^{(k+1)}) - \nabla f(\mathbf{x}^{(k)}), \mathbf{x}^* - \mathbf{x}^{(k+1)} \rangle. \quad (20)$$

The iteration $\mathbf{x}^{(k+1)}$ satisfies the first-order optimality condition for Bregman projection onto the convex set,

$$\langle \nabla f(\mathbf{x}^{(k+1)}) - \nabla f(\mathbf{x}^{(k)}), \mathbf{x} - \mathbf{x}^{(k+1)} \rangle \geq 0, \ \forall \mathbf{x} \in \mathcal{C}_l. \quad (21)$$

Substituting $\mathbf{x} = \mathbf{x}^*$ into Eq. 21 implies that the inner product term in Eq. 20 is nonnegative. Thus,

$$\mathrm{KL}(\mathbf{x}^* \mid \mathbf{x}^{(k)}) \geq \mathrm{KL}(\mathbf{x}^* \mid \mathbf{x}^{(k+1)}) + \mathrm{KL}(\mathbf{x}^{(k+1)} \mid \mathbf{x}^{(k)}). \quad (22)$$

It follows that the sequence $\{\mathrm{KL}(\mathbf{x}^* \mid \mathbf{x}^{(k)})\}$ is non-increasing and bounded below by zero, hence convergent. Summing Eq. 22 over $k$ yields:

$$\sum_{k=0}^{\infty} \mathrm{KL}(\mathbf{x}^{(k+1)} \mid \mathbf{x}^{(k)}) \leq \mathrm{KL}(\mathbf{x}^* \mid \mathbf{x}^{(0)}) < \infty,$$

which implies $\mathrm{KL}\left(\mathbf{x}^{(k+1)} \mid \mathbf{x}^{(k)}\right) \to 0$ as $k \to \infty$. By leveraging the convexity property of the KL-divergence, it can ensure $\left\| \mathbf{x}^{(k+1)} - \mathbf{x}^{(k)} \right\| \to 0$.

Since the sequence $\{\mathbf{x}^{(k)}\}$ is bounded, by the Bolzano-Weierstrass theorem, there exists a convergent subsequence $\{\mathbf{x}^{(k_j)}\} \to \bar{\mathbf{x}}$. By the continuity of the projection operator, we have $\bar{\mathbf{x}} \in \mathcal{C}_l$ for all $l$, and it thus follows that $\bar{\mathbf{x}} \in \mathcal{C}$.

Because the sequence $\{\mathrm{KL}(\mathbf{x}^* \mid \mathbf{x}^{(k)})\}$ converges and $\mathrm{KL}\left(\bar{\mathbf{x}} \mid \mathbf{x}^{(k)}\right) \to 0$, it follows that $\mathbf{x}^{(k)}$ converges asymptotically to $\bar{\mathbf{x}}$. Therefore, $\bar{\mathbf{x}}$ is a solution to Eq. 7, i.e., $\bar{\mathbf{x}} = \mathbf{x}^*$. Moreover, since the optimal solution $\mathbf{x}^*$ is unique, all convergent subsequences must converge to the same limit. Therefore, the entire sequence $\{\mathbf{x}^{(k)}\}$ converges to $\mathbf{x}^*$. $\qquad\square$

**Theorem E.2** (Proposition). *The sequence $\{\mathbf{x}^{(k)}\}$ generated by the Bregman iterative projection method converges superlinearly to the solution $\mathbf{x}^*$.*

*Proof.* Define the iteration function

$$f(\mathbf{x}) = (\mathbf{1}_n - \mathbf{w}_{(l)}) \odot \mathbf{x} + \mathbf{w}_{(l)} \odot \frac{b_{(l)}}{\mathbf{w}_{(l)}^\top \mathbf{x}} \mathbf{x}. \tag{23}$$

Assume that $\lim_{k \to +\infty} \mathbf{x}^{(k)} = \mathbf{x}^*$, then $f(\mathbf{x}^*) = \mathbf{x}^*$ and $\mathbf{w}_{(l)}^\top \mathbf{x}^* = b_{(l)}$.

Perform a Taylor expansion of $\mathbf{x}^{(k+1)} = f(\mathbf{x}^{(k)})$ around $\mathbf{x}^*$:

$$\mathbf{x}^{(k+1)} = f(\mathbf{x}^*) + \nabla f(\mathbf{x}^*)^\top (\mathbf{x}^{(k)} - \mathbf{x}^*) + o(\|\mathbf{x}^{(k)} - \mathbf{x}^*\|_2),$$

where Jacobian matrix $\nabla f(\mathbf{x}^*)^\top$ can be computed as follows:

$$\nabla f(\mathbf{x}^*)^\top = \mathrm{diag}(\mathbf{1}_n - \mathbf{w}_{(l)}) + (\mathbf{w}_{(l)} \odot \mathbf{x}^*) \nabla^\top \left( \frac{b_{(l)}}{\mathbf{w}_{(l)}^\top \mathbf{x}^*} \right) + \frac{b_{(l)}}{\mathbf{w}_{(l)}^\top \mathbf{x}^*} \mathrm{diag}(\mathbf{w}_{(l)})$$

$$= \mathbf{I}_n - \frac{(\mathbf{w}_{(l)} \odot \mathbf{x}^*)}{b_{(l)}} \mathbf{w}_{(l)}^\top. \tag{24}$$

Now, consider the ratio of successive errors in the L2 norm. Let $e^{(k)} = \mathbf{x}^{(k)} - \mathbf{x}^*$. From Eq. 23 and Eq. 24,

$$\frac{\|\mathbf{x}^{(k+1)} - \mathbf{x}^*\|_2}{\|\mathbf{x}^{(k)} - \mathbf{x}^*\|_2} = \frac{\|\nabla f(\mathbf{x}^*)^\top (\mathbf{x}^{(k)} - \mathbf{x}^*) + o(\|e^{(k)}\|_2)\|_2}{\|e^{(k)}\|_2}$$

$$= \frac{\|(\mathbf{I}_n - \frac{(\mathbf{w}_{(l)} \odot \mathbf{x}^*)}{b_{(l)}} \mathbf{w}_{(l)}^\top) e^{(k)} + o(\|e^{(k)}\|_2)\|_2}{\|e^{(k)}\|_2}$$

$$= \frac{\|(\mathbf{I}_n - \frac{(\mathbf{w}_{(l)} \odot \mathbf{x}^*)}{b_{(l)}} \mathbf{w}_{(l)}^\top) e^{(k)}\|_2}{\|e^{(k)}\|_2}.$$

Thus, the squared ratio satisfies

$$\lim_{k \to +\infty} \frac{\|\mathbf{x}^{(k+1)} - \mathbf{x}^*\|_2^2}{\|\mathbf{x}^{(k)} - \mathbf{x}^*\|_2^2} = \lim_{e^{(k)} \to 0} \frac{((\mathbf{I}_n - \frac{(\mathbf{w}_{(l)} \odot \mathbf{x}^*)}{b_{(l)}} \mathbf{w}_{(l)}^\top) e^{(k)})^\top ((\mathbf{I}_n - \frac{(\mathbf{w}_{(l)} \odot \mathbf{x}^*)}{b_{(l)}} \mathbf{w}_{(l)}^\top) e^{(k)})}{e^{(k)T} e^{(k)}} \tag{25}$$

$$= \lim_{e^{(k)} \to 0} \frac{e^{(k)T} (\mathbf{I}_n - \mathbf{w}_{(l)} \frac{(\mathbf{w}_{(l)} \odot \mathbf{x}^*)}{b_{(l)}}) (\mathbf{I}_n - \frac{(\mathbf{w}_{(l)} \odot \mathbf{x}^*)}{b_{(l)}} \mathbf{w}_{(l)}^\top) e^{(k)}}{e^{(k)T} e^{(k)}}. \tag{26}$$

This limit depends only on the direction of $e^{(k)}$, not its magnitude. Define

$$\mathbf{M} = \left( \mathbf{I}_n - \mathbf{w}_{(l)} \frac{(\mathbf{w}_{(l)} \odot \mathbf{x}^*)}{b_{(l)}} \right) \left( \mathbf{I}_n - \frac{(\mathbf{w}_{(l)} \odot \mathbf{x}^*)}{b_{(l)}} \mathbf{w}_{(l)}^\top \right),$$

and Let $\lambda$ be an eigenvalue of $M$ with corresponding eigenvector $\mathbf{p}$. To find all eigenvalues and eigenvectors of $M$, set

$$\mathbf{u} := \mathbf{w}_{(l)} \in \{0, 1\}^n, \quad \mathbf{v} := \frac{\mathbf{w}_{(l)} \odot \mathbf{x}^*}{b_{(l)}}.$$

Then $\mathbf{u}^\top \mathbf{v} = \mathbf{v}^\top \mathbf{u} = 1$.

Denote

$$s := \mathbf{u}^\top \mathbf{u} = \|\mathbf{u}\|^2, \quad \alpha := \mathbf{v}^\top \mathbf{v} = \|\mathbf{v}\|^2.$$

If a vector $\mathbf{z}$ satisfies $\mathbf{u}^\top \mathbf{z} = 0$ and $\mathbf{v}^\top \mathbf{z} = 0$ (i.e., $\mathbf{z}$ is orthogonal to both $\mathbf{u}$ and $\mathbf{v}$), then

$$(\mathbf{I}_n - \mathbf{v}\mathbf{u}^\top)\mathbf{z} = \mathbf{z}, \quad (\mathbf{I}_n - \mathbf{u}\mathbf{v}^\top)\mathbf{z} = \mathbf{z},$$

hence

$$\mathbf{Mz} = (\mathbf{I}_n - \mathbf{uv}^\top)(\mathbf{I}_n - \mathbf{vu}^\top)\mathbf{z} = \mathbf{z}.$$

Thus any vector in $\text{span}(\mathbf{u}, \mathbf{v})$ is an eigenvector with eigenvalue $1$.

Since

$$(\mathbf{I}_n - \mathbf{vu}^\top)\mathbf{v} = \mathbf{v} - (\mathbf{u}^\top\mathbf{v})\mathbf{v} = \mathbf{v} - \mathbf{v} = 0,$$

we have

$$\mathbf{Mv} = 0,$$

so $\mathbf{v}$ is an eigenvector with eigenvalue $0$. Moreover,

$$(\mathbf{I}_n - \mathbf{vu}^\top)\mathbf{u} = \mathbf{u} - (\mathbf{u}^\top\mathbf{u})\mathbf{v} = \mathbf{u} - s\mathbf{v}.$$

Then

$$\begin{aligned}
\mathbf{Mu} &= (\mathbf{I}_n - \mathbf{uv}^\top)(\mathbf{u} - s\mathbf{v}) \\
&= (\mathbf{u} - s\mathbf{v}) - \mathbf{uv}^\top(\mathbf{u} - s\mathbf{v}).
\end{aligned}$$

Since $\mathbf{v}^\top\mathbf{u} = 1$, we have $\mathbf{v}^\top(\mathbf{u} - s\mathbf{v}) = 1 - s\alpha$. Therefore,

$$M\mathbf{u} = \mathbf{u} - s\mathbf{v} - \mathbf{u}(1 - s\alpha) = s\alpha\mathbf{u} - s\mathbf{v}.$$

In the basis $(\mathbf{u}, \mathbf{v})$, consider an eigenvector of the form $\mathbf{u} - k\mathbf{v}$. Then $M(\mathbf{u} - k\mathbf{v}) = \lambda(\mathbf{u} - k\mathbf{v})$ implies $M\mathbf{u} = \lambda(\mathbf{u} - k\mathbf{v})$. Solving this yields the eigenvector $\mathbf{u} - \frac{1}{\alpha}\mathbf{v}$ with eigenvalue $s\alpha$.

By properties of the Rayleigh quotient, the limit in (8) lies between the minimum and maximum eigenvalues of $M$, depending on the direction of $e^{(k)}$. The limit attains the minimum eigenvalue $0$ if and only if $e^{(k)}$ is parallel to the eigenvector corresponding to $0$, i.e., $\mathbf{v} = \frac{\mathbf{w}_{(l)} \odot \mathbf{x}^*}{b_{(l)}}$.

From the iteration formula,

$$\mathbf{x}^{(k+1)} - \mathbf{x}^{(k)} = \left(\mathbf{w}_{(l)} \odot \mathbf{x}^{(k)}\right)\left(\frac{b_{(l)}}{\mathbf{w}_{(l)}^\top\mathbf{x}^{(k)}} - 1\right).$$

Thus $\mathbf{x}^{(k+1)} - \mathbf{x}^{(k)}$ is always parallel to $\mathbf{w}_{(l)} \odot \mathbf{x}^{(k)}$. Since $\lim_{k \to +\infty} \mathbf{x}^{(k)} = \mathbf{x}^*$, for sufficiently large $k$ we have $\mathbf{x}^{(k+1)} - \mathbf{x}^{(k)}$ parallel to $\mathbf{w}_{(l)} \odot \mathbf{x}^*$, and hence $e^{(k)}$ becomes parallel to $\mathbf{v}$. Therefore, in the limit, $e^{(k)}$ is parallel to the eigenvector of $M$ with eigenvalue $0$, and consequently the limitation in Eq. 25 equals $0$. That is

$$\lim_{k \to +\infty} \frac{\|\mathbf{x}^{(k+1)} - \mathbf{x}^*\|_2^2}{\|\mathbf{x}^{(k)} - \mathbf{x}^*\|_2^2} = 0,$$

which implies that the sequence $\{\mathbf{x}^{(k)}\}$ converges superlinearly in the L2 norm.

$\square$

## F. Additional Theoretical Discussions

### F.1. On the Binary Assumption in BLCLayer

The proposed LinConLayer framework consists of two complementary branches:

- **BLCLayer**, which enforces linear constraints with binary coefficients via KL-divergence based Bregman projections;

- **GLCLayer**, which enforces general linear equality constraints via a Euclidean projection with a closed-form solution.

The assumption that the constraint matrix $\mathbf{W}$ is binary is imposed *only* for BLCLayer and is made for technical and efficiency reasons. It is not a fundamental limitation of the overall framework: for non-binary or continuous linear constraints one can simply use GLCLayer, which handles arbitrary real-valued coefficients.

From an application standpoint, binary constraint matrices appear naturally in many structured prediction tasks, especially those with probabilistic outputs and "set selection / counting" structure, such as classification, (partial) matching, and graph-based problems. In these settings the constraints typically encode which entries must be selected or that row/column sums must equal prescribed cardinalities, so a 0–1 structure for $\mathbf{W}$ is a natural modeling choice rather than an artificial restriction.

The binary structure also enables particularly efficient Bregman projections. In BLCLayer each constraint row $w_\ell^\top x = b_\ell$ admits an element-wise update in closed form, so that one Bregman step reduces to simple elementwise multiplications and divisions in the spirit of Sinkhorn iterations. This is crucial for the strong empirical efficiency observed in our graph matching and portfolio experiments.

While BLCLayer is designed for the binary case, several directions suggest how one might extend the KL-based branch to more general coefficient patterns:

- **Row decomposition / reparameterization.** For some constraint rows $w_\ell$, the coefficients can be expressed as a linear combination of a small number of weight groups (e.g., variables sharing the same pattern). Introducing auxiliary variables then allows one to decompose such a row into several 0–1 selection constraints, thereby retaining the fast BLCLayer updates while preserving the continuous information in the original coefficients.

- **One-dimensional convex subproblems under KL.** In the non-binary case, the Bregman projection associated with a single constraint row reduces to a one-dimensional convex optimization problem. This subproblem can be solved numerically, e.g., by a few Newton or bisection iterations, and different rows remain fully parallelizable. This yields an intermediate design point between binary BLCLayer and the fully closed-form GLCLayer, preserving KL geometry while supporting general $\mathbf{W}$.

- **Hybrid designs.** Within the same network, one can use BLCLayer for naturally 0–1 structural constraints (e.g., combinatorial or cardinality constraints) and GLCLayer for general real-valued physical or economic constraints. This hybrid allocation strategy highlights the practical flexibility of LinConLayer when modeling mixed constraint structures.

A full development of these ideas is left for future work; here we include them to clarify the scope of the current design and to indicate promising extensions.

## F.2. Extensions beyond Linear Constraints

The main paper focuses on linear constraints of the form

$$\mathcal{C} = \{\mathbf{x} \in \mathbb{R}^n : \mathbf{W}\mathbf{x} = \mathbf{b}, \mathbf{x} \geq \mathbf{0}\}.$$

Conceptually, the same implicit-layer philosophy could be applied to more general constraint sets, such as quadratic, nonlinear, or logical constraints, but this raises several design and numerical challenges.

As a simple example, consider a quadratic constraint

$$g(\mathbf{x}) \leq \mathbf{0}, \quad g(\mathbf{x}) = \mathbf{x}^\top \mathbf{Q}\mathbf{x} + \mathbf{r}^\top \mathbf{x} + \mathbf{s}.$$

Given an unconstrained network output $\mathbf{y}$, one could in principle enforce $g(\mathbf{x}) \leq 0$ by solving the projection problem

$$\min_{\mathbf{x}} \ D(\mathbf{x}, \mathbf{y}) \quad \text{s.t.} \quad g(\mathbf{x}) \leq 0,$$

where $D$ is a divergence such as KL or a squared Euclidean distance. A straightforward implementation would introduce a penalty parameter $\lambda > 0$ and consider

$$\min_{x} \ D(\mathbf{x}, \mathbf{y}) + \lambda \, \phi(g(\mathbf{x})),$$

for some convex penalty $\phi$, and solve the resulting unconstrained problem by gradient-based iterations.

However, this naive strategy has several drawbacks when used as a neural network layer:

- It introduces additional hyperparameters (penalties, step sizes, stopping criteria), making both training and tuning more cumbersome.

- It typically does not admit a closed-form solution, complicating convergence analysis and the stability of backpropagation.

- It can significantly increase computational cost compared to the simple closed-form or Bregman updates used in our linear setting.

The key challenge in extending LinConLayer beyond linear constraints is therefore to design implicit layers that (i) avoid unnecessary hyperparameters, (ii) admit well-behaved gradients, and (iii) remain computationally efficient. Exploring such generalizations—for instance via more general differentiable convex solvers plugged into the same implicit framework—is an interesting direction for future work, but lies outside the scope of the present paper.

### F.3. Universal Approximation with Projection Layers

A natural question is whether appending a fixed projection layer at the network output reduces the representational power of the model. Here we provide a short argument showing that, for constraint-satisfying target mappings, the projection does not weaken universal approximation properties.

Let $\mathcal{C} = \{\mathbf{x} \in \mathbb{R}^n : \mathbf{W}\mathbf{x} = \mathbf{b}, \mathbf{x} \geq \mathbf{0}\}$ denote the feasible set defined by the linear constraints, and let $\Pi_{\mathcal{C}}$ be the Euclidean projection onto $\mathcal{C}$,

$$\Pi_{\mathcal{C}}(y) = \arg\min_{x \in \mathcal{C}} \|\mathbf{x} - \mathbf{y}\|_2^2.$$

It is well known that $\Pi_{\mathcal{C}}$ is 1-Lipschitz and acts as the identity on $\mathcal{C}$. Consider the class of continuous target mappings

$$\mathcal{F}_{\mathcal{C}} := \{f : \mathcal{X} \to \mathcal{C} \mid f \text{ is continuous}\}.$$

Let $\{g_\theta\}_\theta$ be a standard neural network family (e.g., a multilayer perceptron or a GNN) with the universal approximation property on $\mathcal{X}$: for any continuous $f : \mathcal{X} \to \mathbb{R}^n$ and any $\varepsilon > 0$, there exists $\theta$ such that

$$\sup_{x \in \mathcal{X}} \|g_\theta(\mathbf{x}) - f(\mathbf{x})\|_2 \leq \varepsilon.$$

**Lemma.** *For any $f \in \mathcal{F}_{\mathcal{C}}$ and any $\varepsilon > 0$, there exists a parameter $\theta$ such that*

$$\sup_{x \in \mathcal{X}} \big\|\Pi_{\mathcal{C}}(g_\theta(\mathbf{x})) - f(\mathbf{x})\big\|_2 \leq \varepsilon.$$

**Sketch of proof.** Fix $f \in \mathcal{F}_{\mathcal{C}}$ and $\varepsilon > 0$. By universal approximation of $\{g_\theta\}$, there exists $\theta$ such that

$$\sup_{x \in \mathcal{X}} \|g_\theta(\mathbf{x}) - f(\mathbf{x})\|_2 \leq \varepsilon.$$

Since $f(\mathbf{x}) \in \mathcal{C}$ for all $\mathbf{x}$, we have $\Pi_{\mathcal{C}}(f(\mathbf{x})) = f(\mathbf{x})$. Using the 1-Lipschitz property of Euclidean projection,

$$\|\Pi_{\mathcal{C}}(g_\theta(\mathbf{x})) - f(\mathbf{x})\|_2 = \|\Pi_{\mathcal{C}}(g_\theta(\mathbf{x})) - \Pi_{\mathcal{C}}(f(\mathbf{x}))\|_2 \leq \|g_\theta(\mathbf{x}) - f(\mathbf{x})\|_2 \leq \varepsilon,$$

for all $x \in \mathcal{X}$. Taking the supremum over $\mathcal{X}$ yields the claim.

This shows that, when the target mapping is required to satisfy the linear constraints by design (i.e., $f(\mathcal{X}) \subseteq \mathcal{C}$), composing a universal base network with the projection $\Pi_{\mathcal{C}}$ preserves universal approximation. The projection layer prevents infeasible outputs but does not exclude any constraint-satisfying target functions. This is consistent with universal approximation results for other projection-based architectures and places LinConLayer within the same theoretical landscape.

## Ethics Statement

This paper aims to advance the state of the art in linear constrained machine learning. While the research may entail various societal implications, we do not identify any that warrant specific emphasis in this paper.

## Reproducibility statement

All experimental results in the paper are reproducible, and the implementation code of ParetoRouter/code for reproducing experimental results will be fully open sourced on Github after the paper is accepted.

## LLM Usage Statement

The contribution of LLM in the work proposed in this article is limited to: 1. polishing given written statements; 2. reviewing the syntax for written sentences. We declare that no experimental data was generated/modified by LLM.

