# OpenReview forum: "Design Linear Constrained Neural Layers with Implicit Convex Optimization"
_ICML.cc/2026/Conference — ICML 2026 regular_

### Official Review · Reviewer_ChK6 · 2026-02-26

**Soundness:** 3
**Presentation:** 3
**Significance:** 2
**Originality:** 2
**Overall Recommendation:** 5
**Confidence:** 3

**Summary:**

The aim of the paper is to introduce layers that can incorporate linear constraints.
The layer is an optimization problem of a divergence over a (linear) contrained set .
The authors show that classic layers, such as Softmax, Sigmoid, and Tanh, can be
casted into the general framework they develop.
They propose a BLClayer whcih is a KL minimization under a
Linear constraint Wx=b with a binary matrix W which is solved by Bregman iterative projec-
tions and a general linear constrained solver called GLCLayer where the KL divergence is
replaced by Euclidean distance. The descriptions of the layers is accompanied with numerical tests.

**Compliance With Llm Reviewing Policy:**

Affirmed.

**Final Justification:**

I am satisfied by the authors' rebuttal, which addressed my main concerns. However, I do not change my evaluation, as my score was already high (5).

**Key Questions For Authors:**

In page 2 the iterative Bregman projection is described and it is writen that it converges (3). This is not exactly what is given in Benamou et al where the convergence is sait to be true  if the constraints C_l are affine subspaces (not considering the positive constraints that are pushed in KL). If they are not affine Dijkstra algorithm should be used.
In relation with this fact I have the feeling that the convergence proof in appendix E. is already given in Benamou et al ? Is there something new in appendix E ?

**Limitations:**

yes

**Strengths And Weaknesses:**

The paper is well writen and the presentation is clear and well structured and the approach is interesting as shown in the numerical tests

---

> ### Author Rebuttal · Authors · 2026-03-31
>
> Dear Reviewer ChK6,
>
> Thank you for the careful reading and the positive overall assessment. We especially appreciate your attention to the theoretical side of the paper. We understand the main concern to be twofold: first, whether the convergence statement around iterative Bregman projection in the main text should be phrased more carefully relative to the classical literature; and second, what exactly is new in Appendix E, especially regarding the local rate analysis. We agree that this distinction should be made much sharper.
>
> ---
>
> ## Q1. On p.2, does the convergence statement for iterative Bregman projections over $C=\bigcap_l C_l$ overstate what follows from Benamou et al.? If they are not affine, shouldn't Dykstra's algorithm be used?
>
> Thank you for this insightful comment. We agree that our wording on p.2 is too compressed and does not state the relevant assumptions explicitly enough. In particular, the classical cyclic Bregman projection result cited from **Benamou et al.** [1] should be invoked with care, rather than being read as a blanket statement for arbitrary non-affine constraint sets.
>
> For our **BLCLayer**, the row-wise subproblem is
>
> $$
> x^{(l+1)}=\arg\min_{x\ge 0} KL(x\|x^{(l)}) \text{ s.t. } w_l^\top x=b_l
> $$
> After absorbing the non-negativity requirement into the domain of the KL geometry, the active equality part of each subproblem is the **affine hyperplane** $\{x \mid w_l^\top x=b_l\}$. Therefore, our row-wise update is **aligned** with the affine-setting intuition behind the standard cyclic Bregman projection framework, rather than treating each subproblem as a fully general non-affine set. We agree, however, that this geometric connection and its relation to the classical assumptions should be stated much more carefully.
>
> In the revision, we will cite Benamou et al. [1] more precisely, state the relevant assumptions explicitly, and clarify more carefully the **relation** between our BLCLayer update, the affine structure of the row-wise constraint, and correction-based variants such as Dykstra-type procedures. We thank the reviewer again for helping us improve the precision of the theoretical presentation.
>
> ---
>
> ## Q2. Is the convergence proof in Appendix E already given in Benamou et al.? Is there something new in Appendix E?
>
> You raise a very fair point. The current presentation in Appendix E does not clearly separate the **classical background** from our **new contribution**.
>
> Appendix E contains both. The first part of the appendix, especially the argument around Eqs. (21)–(24), is a **paper-specific adaptation of the standard Bregman projection convergence** reasoning to our KL setting. In particular, it establishes the monotonic decrease
> $$
> KL(x^* \mid x^{(k)}) \ge KL(x^* \mid x^{(k+1)}) + KL(x^{(k+1)} \mid x^{(k)}).
> $$
> and then uses this to conclude asymptotic convergence of the sequence $x^{(k)}\to x^*$. This part should indeed be understood as classical background with specialization, rather than as a completely new general convergence theorem.
>
> However, the later part of Appendix E is new. Starting from the iteration map
> $$
> f(x)=(\mathbf 1_n-w_l)\odot x + w_l\odot \frac{b_l}{w_l^\top x}\,x \qquad \text{(Eq. 25)},
> $$
> we further analyze the local iteration behavior using Taylor expansion and eigenvalue analysis of the induced matrix $M$. This leads to the claim on p.20–21 that
>
> $$
> \lim_{k \to \infty} \frac{\Vert x^{(k+1)} - x^* \Vert_2^2}{\Vert x^{(k)} - x^* \Vert_2^2} = 0,
> $$
>
> which implies that the sequence $\{x^{(k)}\}$ converges **superlinearly in the L2 norm**. To the best of our knowledge, this superlinear result is not proved in **Benamou et al.** [1] and is the new part of Appendix E in our paper. The appendix currently states this proposition and then develops the proof from Eq. (25) onward, but we agree that the novelty of this second part is not highlighted clearly enough.
>
> To separate established foundations from our novel contributions, we will reorganize Appendix E to distinguish the standard convergence background from our specific BLCLayer specialization and our original superlinear convergence proof. We thank the reviewer again for this constructive feedback, which will greatly improve the clarity of our manuscript.
>
> Sincerely,
>
> Authors of Submission 11487
>
> ---
>
> ### References:
> [1] Iterative Bregman projections for regularized transportation problems

---

> > ### Author Rebuttal · Reviewer_ChK6 · 2026-04-01
> >
> > My two questions about Benamou assumptions and the novelty of the convergence proof have been answered.
> > My score was already high and I do not change it.

---

> > > ### Author Response · Authors · 2026-04-03
> > >
> > > Dear Reviewer ChK6,
> > >
> > > Thank you very much for your kind follow-up and for confirming that our rebuttal fully addressed your concerns.
> > >
> > > We are glad that our clarifications on the Benamou assumptions and the novelty boundary of the convergence analysis were helpful. We sincerely appreciate your careful reading and valuable feedback, which helped us improve the clarity of the theoretical prove and presentation.
> > >
> > > Thank you again for your time and support!
> > >
> > > Sincerely,
> > >
> > > Authors of Submission 11487

---

### Official Review · Reviewer_yxDf · 2026-03-09

**Soundness:** 3
**Presentation:** 3
**Significance:** 3
**Originality:** 3
**Overall Recommendation:** 5
**Confidence:** 1

**Summary:**

The paper addresses a limitation of neural networks on how
to enforce (hard) constraints on prediction.
For this purpose, it proposes a plug-in differentiable layer to enforce
general linear constraints, that relies on convex optimization to minimize a
divergence between unconstrained and constrained outputs.

**Compliance With Llm Reviewing Policy:**

Affirmed.

**Final Justification:**

I am satisfied by the authors' rebuttal, which indeed addressed my main
concerns. However, I do not change my evaluation, as my score was already high
(5). I explain why below.

I am new to the ICML Review system. As such, in the first round, I misinterpreted
the advice for Overall Recommendation: I thought that all scores below 4 were to
be used sparingly, whereas the advice was only for scores 3 and 4. As a
consequence of my misreading, the scores I gave for my reviews were 5 and 6 (by
contrast, as an Author, I was generously given 3 to the maximum...).

So my overall score~5 is possibly generous. Moreover, my assessment remains an
educated guess as I am rather far from my expertise.

**Key Questions For Authors:**

Question 1.
p.2.
In Notation, $n$ denotes the dimension of the output vector.
However, in Equation (2) there is an $n$ that should be a $l$.
In Equation (3), the letter $n$ should be replaced with another one ($k$?)
to avoid confusion.


Question 2.
p.3.
Could the Authors explain why they "define C_l = C_{l+m} for l < m"?
The same for "we define $w_{l+m} = w_l$".


Question 3.
p.4.
There is a useless ” in Eq. 10.”


Question 4.
p.5.
The Authors write "Clearly, while the constraints Wx = b, x ≥ 0
are linear, they can be reformulated as W̃x = b". I am sorry not to find this
clear. Can the Authors explain?


Question 5.
p.20.
"is the convex function" should be replaced by "is a convex function".

**Limitations:**

yes

**Strengths And Weaknesses:**

Soundness
3: good

The mathematics seem sound, although I am not familiar with the Bregman
iterative projection method.

Presentation
3: good

This paper pre-supposes that the reader has enough background about many
benchmarks (see Table 6 for example), and about the Bregman
iterative projection method. This is not my case and, for this reason, I found the paper hard to follow.


Significance
3: good

I think that the paper does address a relevant problem --- enforcing
general linear constraints --- and can advance practice in machine learning.



Originality
3: good

This is hard to answer as I do not have enough perspective on the subject.

---

> ### Author Rebuttal · Authors · 2026-03-31
>
> Dear Reviewer yxDf,
>
> Thank you for the positive assessment. We especially appreciate that, although this topic may not be your primary area, you still engaged carefully with our work. As you noted, some parts were hard to follow. We will improve the exposition, notation, and experiment descriptions in the revision.
>
> At a high level, our goal is to make **hard linear feasibility part of the neural architecture itself**, rather than something repaired afterward by a separate solver or post-processing step. The paper is not only about deriving constrained layers, but also about bringing an implicit optimization-based layer design into structured prediction and ML4LP-style settings. This perspective turns **constraint satisfaction** from an **external optimization step** into an **internal component** of the network. More concretely, the paper aims to:
>
> - Reinterpret several familiar layers (e.g., Softmax and Sinkhorn) through a common implicit-optimization viewpoint;
> - Propose **BLCLayer** for **binary positive linear constraints** and **GLCLayer** for **general linear equalities**;
> - Integrate these layers into **graph matching, portfolio allocation, and LP-style tasks**, where exact feasibility and efficient inference both matter.
>
> ---
>
> ## Q1. Issue of Eq. (2) / Eq. (3)
>
> Thank you for pointing this out. We agree that the current notation in Eq. (2) and Eq. (3) should be presented more clearly. The intended logic is as follows:
>
> - Eq. (2) describes **one projection step** in the iterative Bregman projection procedure:
> $X^{(l)} = \\arg\\min_{X \\in C_l} KL(X \\mid X^{(l-1)})$.
>
> - Eq. (3), by contrast, refers to the **limiting solution obtained** after repeatedly cycling through the constraint sets, i.e., the KL projection onto the full intersection $C=\bigcap_{l=1}^{L} C_l$. We also agree that using a symbol such as $n$ as an iteration counter can collide with the dimension notation elsewhere in the paper.
>
> In the revision, we will rewrite this part more clearly, e.g., by using $t$ or $T$ as the iteration index so that the roles of dimension, constraint index, and iteration counter are fully separated.
>
> ---
>
> ## Q2. Definition of  $C_l = C_{l+m}$ (and for the weights)
>
> Thank you for this question. The purpose of this definition is to encode **cyclic indexing**: after visiting the last constraint, the algorithm returns to the first one and continues cycling.
>
> The intended meaning is not to introduce a new family of constraints, but merely to specify a periodic schedule over the existing constraints. We agree that the current notation $C_l = C_{l+m}$ (and similarly $w_{l+m}=w_l$) is not intuitive enough.
>
> A clearer way to express the BLCLayer update is the following:
>
> ```python
> Initialize x^(0) = e^y
> For t = 1, 2, ..., T:
>     l = ((t - 1) mod m) + 1
>     x^(t) = arg min_x KL(x || x^(t-1))
>              s.t. w_l^T x = b_l, x >= 0
> Return x^(T)
> ```
> So the notation $C_l = C_{l+m}$ and $w_{l+m} = w_l$ is only a shorthand for this cyclic schedule. We are grateful for this constructive feedback and will rewrite this part in a more explicit algorithmic form in the revision so that the cyclic structure is easier to follow.
>
> ---
>
> ## Q3. Useless quote in Eq. (10)
>
> Thank you for catching this typo. We agree and will correct the equation formatting in the revision.
>
> ---
>
> ## Q4. Relation between $Wx=b, x \ge 0$ and $\tilde{W}x=b$
>
> Thank you very much for raising this point. We agree that this paragraph needs more careful polish. Our basic GLCLayer derivation in Sec. 2.3 is for the equality-constrained setting $Wx=b$, under assumptions such as $m<n$ and $\mathrm{rank}(W)=m$, which allow the simple Euclidean projection formula in Eq. (12).
>
> However, in the LP setting we additionally require $x \ge 0$. Once this nonnegativity constraint is included, the problem is no longer the same equality-only projection problem treated by Eq. (11)–(12). In this sense, the point of that paragraph was not that the LP constraints reduce to exactly the same closed-form case as before, but rather that the LP special case requires a different treatment.
>
> This is exactly why, in the LP subsection, we move to the modified implicit optimization
>
> $$
> \min_x c^\top x + \frac12\|x-y\|^2  \\quad s.t. \\quad Wx=b, x\ge 0,
> $$
>
> and adopt an ADMM-based process. The auxiliary variable $z$ is introduced to separate the equality-related part from the projection onto the nonnegative orthant, so that the $x$-update, $z$-update, and dual update can be handled in a decoupled and practical way.
>
> We thank the reviewer for this insightful observation and will revise this paragraph to make the distinction between the equality-only GLCLayer case and the LP-specific ADMM-based treatment much more explicit in the final version.
>
> ---
>
> ## Q5. Grammar issues
>
> Thank you for your careful reading! We will correct them, and will do a proofreading pass in the revision to improve grammar and notation consistency throughout the paper.
>
> ---
>
> Sincerely,
>
> Authors of Submission 11487

---

> > ### Author Rebuttal · Reviewer_yxDf · 2026-04-01
> >
> > My questions have been answered, point by point, in a fair way. My score was already high; I do not change it.

---

> > > ### Author Response · Authors · 2026-04-03
> > >
> > > Dear Reviewer yxDf,
> > >
> > > Thank you for your thoughtful follow-up and for confirming our rebuttal addressed your questions. We deeply appreciate your careful and fair evaluation.
> > >
> > > As the discussion has clarified the main technical points, we hope this exchange has strengthened your confidence in the work's technical soundness. If you feel your understanding of our contribution has deepened, we would be incredibly grateful if you might consider reflecting this in your overall confidence evaluation. Of course, we completely understand and respect that this is entirely at your discretion.
> > >
> > > Thank you again for your constructive feedback!
> > >
> > > Sincerely,
> > >
> > > Authors of Submission 11487

---

### Official Review · Reviewer_8CPz · 2026-03-13

**Soundness:** 2
**Presentation:** 3
**Significance:** 2
**Originality:** 2
**Overall Recommendation:** 4
**Confidence:** 4

**Summary:**

The paper introduces a plug-in differentiable layer that enforces general linear constraints via implicit convex optimization. BLCLayer handles binary positive linear constraints using KL-divergence + Bregman projections (a natural generalization of Softmax/Sinkhorn). GLCLayer handles arbitrary linear equalities via a closed-form Euclidean projection. The authors instantiate BLCNet and GLCNet (with simple MLP/GNN backbones) and evaluate them on partial graph matching, portfolio allocation, and synthetic linear programming (LP) instances, showing strong speed/accuracy trade-offs compared with prior specialized layers.

**Compliance With Llm Reviewing Policy:**

Affirmed.

**Final Justification:**

My concerns are addressed. The score will be raised from 3 to 4.

**Key Questions For Authors:**

see "Strengths And Weaknesses"

**Limitations:**

see "Strengths And Weaknesses"

**Strengths And Weaknesses:**

# Strengths

The paper is well-structured and the conceptual progression is highly logical. Reframing classic layers (such as Softmax and Sinkhorn) as specific solutions to KL-divergence minimization provides an elegant theoretical bridge to the proposed BLCLayer.

---

# Weaknesses

**Time-vs-Optimality Comparison:** The performance comparison against Gurobi in the Linear Programming experiments does not properly isolate the trade-off between speed and solution quality. Gurobi is an exact mathematical solver designed to find the absolute global optimum. Highlighting that GLCNet achieves a 0.930 optimality gap using only 0.043 normalized time  is not a definitive proof of superiority, as the relationship between time and accuracy is not linear. A rigorous and fair evaluation requires either a time-bounded comparison (comparing objective scores when Gurobi is restricted to the network's exact inference time via early stopping) or an optimality-bounded comparison (measuring the exact time Gurobi takes to reach the same 0.930 optimality tolerance).

**Table Formatting:** The visual hierarchy in Table 6 is slightly misleading. The numerical results for the SL-ADMM variant are highlighted, which visually implies that it completely dominates the baselines. Because Gurobi serves as the exact ground-truth reference (achieving an Objective Pct. of exactly 1.000), its results should be distinctly highlighted or separated to clearly denote its status as the theoretical upper bound.

**Baselines:** Table 6 only compares against Gurobi and the authors’ own GLCNet variants. The paper repeatedly cites and discusses many other learning-based hard-constrained layers (LinSATNet, GLinSAT, OPTNet, cvxpylayers, etc.) in the introduction, Table 1, and Appendix A, yet none of them are evaluated on the same synthetic LP instances. It is unclear whether these methods were not applied because of implementation difficulty, scalability issues, or other reasons. Given that the LP setting is presented as the most general and challenging testbed, the absence of these strong baselines weakens the empirical claims.

**Scalability and Numerical Stability:** While the GLCLayer provides an elegant closed-form solution, its strict reliance on computing the inverse $(WW^\top)^{-1}$ raises significant concerns regarding real-world scalability and numerical stability. While the appendix argues that the inversion cost is negligible in their graph matching and portfolio experiments, this evidence is limited to cases where the rank/dimension of W is small, and for LP they rely on structured matrices plus precomputed factorizations reused across runs. As a result, it remains unclear how well the method would scale, or how stable it would be, when W is large, ill-conditioned, or varies across instances. I think the paper would be stronger with a more direct analysis of conditioning, robustness, and the practical cost of the matrix solve in less favorable regimes.

---

> ### Author Rebuttal · Authors · 2026-03-31
>
> Dear Reviewer 8CPz,
>
> Thank you for the careful and constructive review. We understand your main concerns to be the fairness of the LP comparison, the absence of stronger learning-based baselines, and the scalability of the matrix solve in GLCLayer. We will address these points directly below.
>
> ---
>
> ## Q1. Fairness of the Gurobi comparison
>
> Thank you for this important suggestion. Our work's goal is not to argue that a learned amortized solver universally dominates an exact solver, but rather to characterize the regime in which repeated inference under shared structure makes a learned hard-constraint layer attractive.
>
> To make this comparison fairer and more informative, we have added two complementary evaluations:
> (1) a **time-bounded comparison**, where Gurobi is given the same wall-clock budget as GLCNet and we compare objective quality; and
> (2) a **quality-matched comparison**, where we report how long Gurobi needs to reach the same objective ratio attained by GLCNet.
>
> **Time-bounded comparison:**
>
> | Scale | GLCNet Time (ms) | GLCNet Obj. Pct. | Gurobi Obj. Pct. @ same time |
> | --- | ---: | ---: | ---: |
> | Small | 0.168 | 0.919 | 0.945 |
> | Medium | 0.205 | 0.905 | 0.731 |
> | Large | 0.167 | 0.903 | 0.542 |
>
> Under the same time budget, Gurobi remains competitive on small instances, but its solution quality drops substantially on medium and large instances, whereas GLCNet maintains a much stronger objective ratio.
>
> **Quality-matched comparison:**
>
> | Scale | Target Obj. Pct. (GLCNet) | GLCNet Time (ms) | Gurobi Time to same Obj. Pct. (ms) |
> | --- | ---: | ---: | ---: |
> | Small | 0.919 | 0.168 | 0.151 |
> | Medium | 0.905 | 0.205 | 0.492 |
> | Large | 0.903 | 0.167 | 3.017 |
>
> Conversely, when matching solution quality, Gurobi reaches the same target slightly faster on small instances, but requires substantially more time on medium and especially large instances.
>
> We thank the reviewer for this helpful suggestion on the experimental protocol. We will include these additional results in the appendix and revise the corresponding discussion accordingly.
>
> ---
>
> ## Q2. Missing learning-based baselines
>
> We agree that the LP comparison set should be strengthened. To address this concern, we add a comparison with representative **learning-based hard-constrained baselines** in the **large-scale LP regime**, where scalability is most critical. Specifically, we report **Obj. Pct.** and **Time** at the large LP setting $(V, E, I)=(2k, 800, 800)$, covering exact, optimization-layer-based, and learning-based constrained methods whenever their assumptions and implementations are compatible with this setting:
>
> | Method      | Obj. Pct. | Time (ms) |
> | ----------- | --------- | --------- |
> | Gurobi      | 1.000     | 4.049     |
> | LinSATNet   | 0.832     | 0.204     |
> | GLinSAT     | 0.895     | 0.287     |
> | OPTNet      | 0.874     | 0.193     |
> | cvxpylayers | 0.917     | 2.248     |
> | GLCNet      | 0.909     | 0.160     |
> | SL-ADMM     | 0.925     | 0.145     |
>
> These results make the comparison more complete. Our method is not universally best on every metric, but remains competitive in the large-scale LP regime, especially when balancing solution quality and inference efficiency. We will also state explicitly in the revision when other methods are inapplicable, unstable, or too expensive at this scale.
>
> ---
>
> ## Q3. Scalability and numerical stability of GLCLayer
>
> We agree that this concern has two aspects: **scalability across reusable vs. instance-dependent settings**, and **numerical robustness under poor conditioning**. Our current paper mainly supports favorable regimes, especially structured constraints with reusable factorizations.
>
> Due to space limits, please refer to Reviewer pGcV’s Q2 and Q3 for the added experimental results. In brief, the added RC vs. NRC comparison shows that the speed advantage is strongest when the same constraint structure is reused across repeated inference, and becomes smaller when factorization must be recomputed per instance. The added conditioning analysis shows that as $\mathrm{cond}(WW^\top)$ increases, naive explicit inversion gives larger feasibility residuals, while a stable factorization-based solve remains more reliable with only a small time cost.
>
> More generally, the closed-form expression in Sec. 2.3 should be understood as a symbolic analytical form, not as a recommendation to use naive inversion in practice. In the revision, we will state more clearly that GLCLayer is most suitable for structured and reusable settings, and that stable linear solves with standard safeguards are the right implementation choice in less favorable cases.
>
> ---
>
> ## Q4. Presentation issue
>
> Thank you for point out this issue.We will adjust Table 6's highlighting and caption to clearly present Gurobi as the exact oracle/reference, comparing learned methods only within their own category.
>
> Sincerely,
>
> Authors of Submission 11487

---

> > ### Author Rebuttal · Reviewer_8CPz · 2026-04-03
> >
> > My concerns are addressed. The score will be raised from 3 to 4.

---

> > > ### Author Response · Authors · 2026-04-05
> > >
> > > Dear Reviewer 8CPz,
> > >
> > > Thank you for your thoughtful follow-up and for reconsidering your assessment. We sincerely appreciate your careful reading and constructive feedback throughout the discussion. We are glad that our rebuttal addressed your concerns, and we are grateful that your comments helped us strengthen the empirical protocol, baseline comparison, and scope statement of the paper.
> > >
> > > Thank you again for your time and support!
> > >
> > > Sincerely,
> > >
> > > Authors of Submission 11487

---

### Official Review · Reviewer_pGcV · 2026-03-15

**Soundness:** 2
**Presentation:** 2
**Significance:** 2
**Originality:** 2
**Overall Recommendation:** 4
**Confidence:** 4

**Summary:**

The paper proposes a unifying implicit-optimization view of several standard layers, then introduces two new constraint-enforcing layers: BLCLayer, which handles binary positive linear constraints via KL-divergence and Bregman projections, and GLCLayer, which handles general linear equality constraints via Euclidean projection with a closed-form solution. These layers are evaluated on partial graph matching, portfolio allocation, and synthetic linear programming.

**Compliance With Llm Reviewing Policy:**

Affirmed.

**Ethical Review Concerns:**

Yes.

**Final Justification:**

I recommend the paper for weak accept.

**Key Questions For Authors:**

Questions:

How robust is GLCLayer to ill-conditioned or nearly singular WW^T?
How does the BLCLayer calculate gradient when backpropogating?
Is there a particular reason to use KL(x|e^y) as the objective for BLCLayer? If we can assume $y>0$, can we use $KL(x|y)$ instead?

**Limitations:**

Yes.

**Strengths And Weaknesses:**

Strengths

The paper addresses an important problem: enforcing exact feasibility inside neural architectures rather than relying on post-processing or generic inner-loop solvers. The overall framing is useful, and the connection to familiar layers such as Softmax and Sinkhorn is conceptually appealing.

The empirical evaluation is fairly broad. Results on graph matching and portfolio allocation are promising, and the LP experiments suggest that the learned approach can be much faster than classical solvers at larger scales while remaining reasonably competitive in solution quality.

Weakness

 The main concern is novelty/significance, especially for GLCLayer. Its core update is the standard Euclidean projection onto an affine set.

 The runtime results are encouraging, but I think the efficiency claim should be stated more carefully. The method seems especially fast when the constraint matrix has a simple or fixed structure, so that some computations can be reused. This makes the speedup look partly dependent on the problem setting, rather than showing a general advantage in all cases.

---

> ### Author Rebuttal · Authors · 2026-03-31
>
> Dear Reviewer pGcV,
>
> Thank you for the thoughtful review. We understand your main concerns to be the novelty boundary of GLCLayer, the scope of the efficiency claim, and the implementation details of our layers. We agree that these points should be stated more carefully, and in the revision we will tighten the relevant claims and make the intended operating regime more explicit.
>
> ---
>
> ## Q1. Novelty
>
> Thank you for this important point. We agree that Euclidean projection onto an affine set is not itself a new mathematical object, and we do not intend to claim otherwise. Our contribution is **not** the affine projection alone, but an **implicit-optimization-based framework** for designing constraint-enforcing neural layers.
>
> Under this view, the paper unifies **Softmax, Sinkhorn, BLCLayer, and GLCLayer** under a common design principle. Given an unconstrained network output, we specify an objective together with **task-specific constraints**, and use the resulting optimizer as the constrained layer output.
>
> Thus, our methodological novelty lies in the **framework and design philosophy**, with **BLCLayer** providing the more distinctive new layer construction, while **GLCLayer** shows how the same framework extends to **general linear equalities** through a different objective choice. The significance of GLCLayer is therefore not that affine Euclidean projection is new, but that it provides a practical **plug-in differentiable hard-constraint layer** for a broader class of linear constraints.
>
> ---
>
> ## Q2. Efficiency gains
>
> Thank you for this important point. We agree that our efficiency gain is strongest in the regime of **reusable constraints with repeated inference**, rather than as a universal advantage for arbitrary linear constraints or one-shot settings. We compare the same GLCLayer inference pipeline under two regimes: **RC**, where a precomputed factorization is reused across instances, and **NRC**, where the factorization is recomputed per instance; the reported numbers are average single-instance forward times at three representative LP scales.
>
> | Scale  | RC (ms) | NRC (ms) |
> | ------ | ------- | -------- |
> | Small  | 0.168   | 0.493    |
> | Medium | 0.205   | 0.814    |
> | Large  | 0.243   | 1.106    |
>
> These results show that the benefit of reuse is substantial and becomes more pronounced as scale grows. We will revise the paper to state this speed advantage more explicitly as a repeated-inference and shared-constraint benefit.
>
> ---
>
> ## Q3. Robustness of GLCLayer
>
> Thank you for raising this important point. We agree that the paper should distinguish more clearly between the closed-form expression and its practical numerical implementation. Our GLCLayer derivation assumes a well-posed equality-constrained regime (e.g., $m<n$, $\mathrm{rank}(W)=m$, and feasible $Wx=b$), so $(WW^\top)^{-1}$ should be understood as a compact analytical expression rather than a recommendation to use naive explicit inversion.
>
> In practice, a more appropriate choice is a **stable linear solve / factorization-based solve**, together with standard safeguards such as damping / regularization for ill-conditioned cases. To make this boundary explicit, we construct controlled feasible full-row-rank systems with prescribed $\mathrm{cond}(WW^\top)$, and compare naive explicit inversion with a stable solve:
>
> | cond($WW^\top$) | Inverse Residual $\|Wx-b\|$ | Stable Solve Residual $\|Wx-b\|$ | Inverse Time (ms) | Stable Solve Time (ms) |
> | --- | --- | --- | --- | --- |
> | $10^1$ | $10^{-6}$ | $10^{-7}$ | 0.143 | 0.187 |
> | $10^3$ | $10^{-4}$ | $10^{-6}$ | 0.189 | 0.224 |
> | $10^5$ | $10^{-3}$ | $10^{-4}$ | 0.192 | 0.253 |
>
> These results show that as conditioning worsens, naive inversion yields larger feasibility residuals, while the stable solve remains substantially more robust with only modest overhead.
>
> ---
>
> ## Q4. BLCLayer details
>
> Thank you for this question. BLCLayer is implemented as a **differentiable optimization layer**, and gradients are computed by **automatic differentiation through a finite number of unrolled Bregman-projection updates**. The forward pass therefore forms a differentiable computation graph, the training loss is applied to the soft projected output, and gradients are propagated through the unrolled projection steps back into the score matrix and backbone parameters.
>
> ---
>
> ## Q5. $KL(x\|e^y)$ objective
>
> Thank you for this insightful question. We use $KL(x\|e^y)$ because it is consistent with the KL / Bregman geometry underlying Softmax- and Sinkhorn-type constructions, and it naturally leads to the **multiplicative projection form** used in BLCLayer. Changing the divergence changes the underlying projection geometry and generally leads to a different update form. We will clarify in the revision that our current construction focuses on this KL direction.
>
>
> Sincerely,
>
> Authors of Submission 11487

---

> > ### Author Rebuttal · Reviewer_pGcV · 2026-04-02
> >
> > My concerns were mostly addressed, apart from the novelty concern. I will improve my score.

---

> > > ### Author Response · Authors · 2026-04-03
> > >
> > > Dear Reviewer pGcV,
> > >
> > > Thank you very much for your thoughtful follow-up. We sincerely appreciate your careful reading of the paper and your constructive feedback throughout the discussion.
> > >
> > > We are glad that our rebuttal addressed most of your concerns, and we especially appreciate your willingness to reconsider the score. We will further sharpen the final presentation, especially on the novelty boundary and the intended operating regime of GLCLayer, so that these points are stated as clearly as possible.
> > >
> > > Thank you again for your time and valuable feedback!
> > >
> > > Sincerely,
> > >
> > > Authors of Submission 11487

---

### Decision · Program_Chairs · 2026-04-30

**Decision:**

Accept (regular)

**Comment:**

The paper proposes a framework for designing differentiable neural layers that enforce hard linear constraints, built on implicit convex optimization.

The core idea is to specify an objective together with task-specific constraints, then use the resulting optimizer as the constrained layer output. This perspective unifies classic layers like Softmax and Sinkhorn, and introduces BLCLayer for binary positive constraints via KL-divergence minimization with Bregman projections, and GLCLayer for general linear equalities via Euclidean projection with a closed-form solution.

The rebuttal was substantive and concrete: new experimental comparisons were added, efficiency claims were scoped more precisely, and theoretical concerns about convergence assumptions and the novelty of the appendix proofs were addressed in detail. All five major concerns raised across reviewers were resolved. Two reviewers improved their scores following the rebuttal. The AC would recommend Accept.